# Time-Varying Aeroelastic Modeling and Analysis of a Rapidly Morphing Wing

**Liqi Zhang and Yonghui Zhao \***

State Key Laboratory of Mechanics and Control of Mechanical Structures, Nanjing University of Aeronautics and Astronautics, Nanjing 210016, China
\* Correspondence: zyhae@nuaa.edu.cn

**Abstract:** Advanced rotational variable-swept missile wings require the ability to rapidly deploy, retract and reach the designated position. Therefore, the establishment of an effective time-varying aeroelastic model of a rotating missile wing is the prerequisite for performing transient response analysis during the rapid morphing process. In this paper, the finite element model of the wing at the fixed configuration is combined with the floating frame method to describe the small elastic deformations and large rigid-body displacements of the wing, respectively. Combining the structural dynamic model with the supersonic piston theory, a nonlinear and time-varying aeroelastic model of a missile wing undergoing the rapid morphing process is established. A method for the real-time determination of the time-varying lifting surface during morphing is discussed. Based on the proposed aeroelastic equations of motion, the flutter characteristics of the wing at different sweep angles are obtained. The influences of the actuator spring constant, the damping ratio during the morphing and the post-lock stages, as well as the velocity quadratic term in the aeroelastic equations, on the transient responses of the system are studied. The simulation results show that the flutter characteristics of the wing are greatly influenced by the sweep angle. Moreover, the jumping phenomenon in flutter speed due to the switching of flutter modes is found with the increase of the sweep angle. The morphing simulations demonstrate that the transient aeroelastic responses mainly occur in the post-lock stage, so much more attention needs to be focused on the post-lock vibrations. In addition, under the given simulation parameters, the nonlinear quadratic velocity term has little effect on the transient responses of the system. This study provides an efficient method for predicting the transient aeroelastic responses of a rotational variable swept wing.

**Keywords:** variable-swept wing; floating frame of the reference formulation; rapidly morphing; time-varying aeroelastic system





## 1. Introduction

Cruise missiles take off from the ground or carrier platforms, fly in the atmosphere and attack various targets. Their flight conditions (altitude, flight Mach number, etc.) will change dramatically. It is difficult for the flight vehicle with a fixed configuration to adapt to such a wide range of changes in flight parameters. Therefore, it is impossible to always maintain an excellent flight performance within the whole flight envelope. One solution is to design aircraft wings that can change shapes in flight such that a single aircraft can optimally perform multiple missions [1]. In summary, the main objective of the morphing design is to resolve the conflicts that arise between high-speed and low-speed flight or to control the flight in different speed regimes [2].

Up to now, various morphing configurations have been proposed, which can be categorized according to the geometric changes to the parameters such as the sweep angle, camber, twist, span and dihedral [3]. For instance, inspired by flying birds, the variable swept aircraft became an earlier batch of flying machines that had been put into production for military use. The earlier morphing aircrafts included the Pterodactyl IV demonstrator,

F-14 fighter, Tornado fighter, Tu-160 bomber, etc. All of these aircrafts achieved an improved flight performance by actively changing the sweep angle. However, after the 1970s, almost all the variable swept aircraft projects were cancelled due to the increased aircraft weight and the structural complexity. In 2003, the U.S. Defense Advanced Research Projects Agency (DAPRA) launched the Morphing Aircraft Structures (MAS) program. Three contractors of this program, Lockheed Martin, NextGen Aeronautics, and Raytheon Missile Systems, proposed the concepts of a folding wing, a flexible skin morphing wing, and a telescoping wing, respectively [4]. After that, the morphing aircraft has regained attention within the aerospace community. At present, the research of morphing aircraft mainly focuses on materials or shape morphing mechanisms and aerodynamic design [5–8].

Li et al. [9] designed different aerodynamic shapes of the variable swept wing and completed the aerodynamic characteristics analysis. They proved the super adaptive ability of the morphing technology. Xia [10] designed the optimal sweep angle of a variable swept Firebee drone for variable cruise missions. Chen et al. [11] proposed an efficient offline trajectory planning method for the morphing aircraft referring to the AGM-158 airfield missile. Furthermore, Dai et al. [12,13] constructed a dynamic model for the variable swept wing, the aerodynamic performance and the impact of the additional forces and moment resulting from morphing motions were studied, but they focused on the flight performance rather than the aeroelastic characteristics. In addition to the variable swept wing, there are some other popular morphing types, such as the folding wing, variable-camber wing and variable-span wing. Zhao and Hu [14] developed a set of differential-algebraic equations to predict the transient responses of the wing during the folding process, which can be applied to both the slow- and fast-varying processes of the folding wing. Zhao [15] studied an adaptive variable-camber wing experimentally and numerically, and they found that the flight efficiency increased by 14.1% compared to the traditional fixed-wing.

It is noted that the aeroelastic effects have significant influence on the design and flight performance of the morphing aircraft. In general, the aircraft will experience great changes in the structure and aerodynamic forces during the morphing process, which will lead to a remarkable change in the aeroelastic characteristics of the system [16]. The aeroelastic problems of the morphing wings have been studied over the past decade. A variety of modeling and analysis methods were developed to investigate the aeroelastic behaviors of the morphing wing. Li [17] analyzed the stability of a variable span wing, whose simulation results indicate that the proposed morphing law could accomplish the flutter suppression.

The aeroelastic model of the morphing wing varies with the variation of the morphing parameters (folding angle, sweep angle, span, etc.). Obviously, the parameter-varying modeling method is the key to the aeroelastic analysis of the morphing wing. In fact, many morphing systems exhibit dynamics that can be reasonably described by Linear Parameter-Varying (LPV)[18] models. The LPV models can be used to represent the dynamic characteristics of a linear parameterized system online and to characterize nonlinear systems from a series of local linear models. Generally, there exist two approaches for LPV modeling: the global approaches [19–21] and the local approaches [22,23]. The global approaches can used to predict responses produced by global parameters (scheduling parameter). However, it is based on the assumption that it is possible to perform a global identification experiment by exciting the system while the scheduling parameters are persistently changing the system dynamics. This assumption, obviously, is invalid for most of the flight vehicles [24]. The local approaches use the time-frozen assumption to generate a set of local Linear Time-Invariant (LTI) models at different fixed parameters. Additionally, then the overall LPV system can be obtained by interpolating these LTI models. Since the state-space representation is not unique, these local LTI models are in general not given in a consistent form. They must be represented in a consistent state-space form before interpolating [25–27]. It is an inherent limitation of the local approaches that only the static parameter-dependent part can be modeled [28], since time variation of the scheduling parameter is not considered in the modeling. Hence, strictly speaking, the local approaches are only applicable to the system with slowly varying parameters.

For a rapid morphing process, the local approaches are not suitable for understanding system dynamic behaviors due to the time-varying nature of the system. It is necessary to develop the time-varying modeling methods to account for the effects produced by the rapid morphing process. Selitrennik [29] developed a new approach for the computational fluid dynamics-based aeroelastic simulation of rapid morphing flight vehicles, in which the fictitious-mass substructure synthesis method was used to obtain the structural equations of motion, and the centrifugal forces were applied on the mass elements to account for the stiffening effect caused by the rotating motion. However, the high-fidelity morphing simulation is very time consuming.

To improve the flight performance, the variable-swept wings can be integrated into the cruise missiles, such as the AGM158 cruise missile series and BGM-109 cruise missiles in service. It is predictable that this kind of variable-swept morphing capability can also be integrated into the supersonic vehicles to perform particular tasks. In such morphing wing applications, it is required that the missile wings have the ability to deploy, retract, or reach the designated position rapidly. In this case, the slowly time-varying hypothesis does not hold true. The goal of this paper is to develop a set of nonlinear, time-varying aeroelastic equations to predict the transient aeroelastic responses of a variable-swept wing during the rapid morphing process. The proposed modeling method is computationally efficient, and can be applied to both slow and fast time-varying processes of the variable-swept wing during the morphing process. The effectiveness of the proposed method was verified through numerical simulations.

## 2. Aeroelastic Modeling of the Rotating Variable Swept Wing

### 2.1. Description of the Motion of the Wing

Figure 1 illustrates the motion of a variable swept wing in three-dimensional space. The wing can be rotated about the original point $o$ to change the sweep angle. The transient responses of the wing during morphing can be regarded as the small elastic vibrations superimposed on a large-scale overall rigid body motion. Therefore, the floating frame method [30] may be the best choice for solving the present problem. In this method, the configuration of the deformable wing is identified by using reference and floating frames. The reference frame $oxyz$ in Figure 1a defines the location and orientation of the wing. The floating frame $ox_wy_wz_w$ is used to describe the small deformations of the wing with respect to the reference frame. Assume that the axes of these two coordinate systems are initially parallel. The wing can rotate around the $z_w$ axis with an angle $\theta$ (the clockwise direction is positive rotation) until it reaches the maximum sweep angle $\theta_{max}$, as shown in Figure 1b. The symbol $U_\infty$ in the figure represents the incoming flow speed. It should be emphasized that the angular velocity of the rotating wing is not necessarily constant, but can be time-varying during the morphing process.

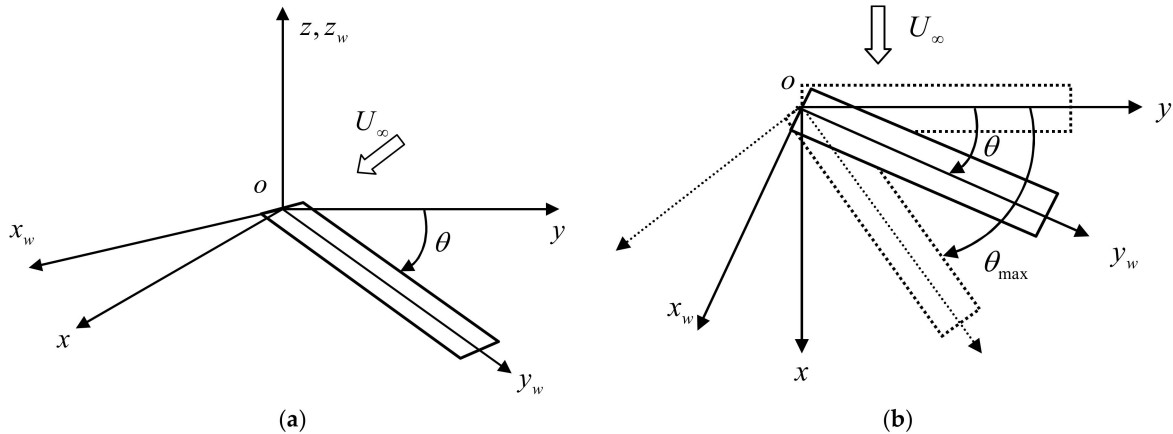

(a)          (b)

**Figure 1.** Schematic of a variable swept wing and the coordinate frames. (**a**) Stereogram description. (**b**) Planform description.

### 2.2. Structural Dynamic Modeling of a Variable Swept Wing

The main problem to be solved in the structural dynamic modeling is to integrate the finite element model (FEM) established by commercial software (e.g., MSC.Nastran) into the floating frame method. Note that the construction of the time-varying aeroelastic equations requires the mass, stiffness and modal matrices of the wing. Therefore, the FEM of the wing should be constructed firstly in the floating frame at the equilibrium configuration. Moreover, in order to facilitate the use of the FEM in the floating frame method, the lumped mass matrix must be specified during the FEM modeling process of the wing.

It is assumed that the FEM of the wing in the floating frame has been established. During the rotation of the wing, as shown in Figure 1a, the instantaneous global position $\boldsymbol{r}_p(t)$ of an arbitrary finite element node $p$ can be written as

$$\boldsymbol{r}_p(\theta, t) = \boldsymbol{A}(\theta)(\overline{\boldsymbol{u}}_{wp} + \boldsymbol{S}_{wp}\boldsymbol{q}_w(t)) \tag{1}$$

where

$$\overline{\boldsymbol{u}}_{wp} = \begin{Bmatrix} x_{wp} \\ y_{wp} \\ z_{wp} \end{Bmatrix} \tag{2}$$

$$\boldsymbol{S}_{wp} = \begin{bmatrix} \phi_{wx,1}(x_{wp}, y_{wp}, z_{wp}) & \phi_{wx,2}(x_{wp}, y_{wp}, z_{wp}) & \phi_{wx,n}(x_{wp}, y_{wp}, z_{wp}) \\ \phi_{wy,1}(x_{wp}, y_{wp}, z_{wp}) & \phi_{wy,2}(x_{wp}, y_{wp}, z_{wp}) & \phi_{wy,n}(x_{wp}, y_{wp}, z_{wp}) \\ \phi_{wz,1}(x_{wp}, y_{wp}, z_{wp}) & \phi_{wz,2}(x_{wp}, y_{wp}, z_{wp}) & \phi_{wz,n}(x_{wp}, y_{wp}, z_{wp}) \end{bmatrix}_{3 \times n} \tag{3}$$

$$\boldsymbol{u}_{wp}(t) = \begin{Bmatrix} u_{wx}(t) \\ u_{wy}(t) \\ u_{wz}(t) \end{Bmatrix} = \boldsymbol{S}_{wp}\boldsymbol{q}(t), \ \boldsymbol{q}_w(t) = \begin{Bmatrix} q_{w1}(t) \\ q_{w2}(t) \\ \vdots \\ q_{wn}(t) \end{Bmatrix} \tag{4}$$

$\overline{\boldsymbol{u}}_{wp}$ is the position vector of the node $p$ in the floating frame $ox_w y_w z_w$. $\boldsymbol{S}_{wp}$ is the matrix related to the modal shapes at the node $p$. $\phi_{wx,i}$, $\phi_{wy,i}$ and $\phi_{wz,i}$, $i = 1, 2, \cdots, n$, are the values of the $i$-th mode at the node $p$ in $x_w$, $y_w$ and $z_w$ directions, respectively, which can be obtained by the normal mode analysis of the FEM. $\boldsymbol{q}_w(t)$ is the modal coordinate vector describing the elastic deformation of the wing, and $n$ is the number of the retained modes. $\boldsymbol{A}$ represents the transformation matrix between the floating frame $ox_w y_w z_w$ and the global frame $oxyz$, given by

$$\boldsymbol{A}(\theta) = \begin{bmatrix} \cos\theta & \sin\theta & 0 \\ -\sin\theta & \cos\theta & 0 \\ 0 & 0 & 1 \end{bmatrix} \tag{5}$$

The velocity vector at the finite element node $p$ can be obtained by differentiating the position vector with respect to the time, given by

$$\dot{\boldsymbol{r}}_p = \begin{bmatrix} \boldsymbol{A}_\theta(\theta)(\overline{\boldsymbol{u}}_{wp} + \boldsymbol{S}_{wp}\boldsymbol{q}_w(t)) & \boldsymbol{A}(\theta)\boldsymbol{S}_{wp} \end{bmatrix} \begin{Bmatrix} \dot{\theta} \\ \dot{\boldsymbol{q}}_w \end{Bmatrix} \tag{6}$$

where

$$\boldsymbol{A}_\theta(\theta) = \begin{bmatrix} -\sin\theta & \cos\theta & 0 \\ -\cos\theta & -\sin\theta & 0 \\ 0 & 0 & 0 \end{bmatrix} \tag{7}$$

Thus, the kinetic energy $T_w$ of the rotating wing can be written as

$$T_w = \frac{1}{2}\sum_{p=1}^{n_w} m_p(\dot{\boldsymbol{r}}_p)^{\mathrm{T}}\dot{\boldsymbol{r}}_p = \frac{1}{2}\dot{\boldsymbol{q}}^{\mathrm{T}}\boldsymbol{M}_w(\boldsymbol{q})\dot{\boldsymbol{q}} = \frac{1}{2}\begin{Bmatrix}\dot{\theta}\\\dot{\boldsymbol{q}}_w\end{Bmatrix}^{\mathrm{T}}\begin{bmatrix}m_{\theta\theta} & m_{\theta q}\\m_{q\theta} & m_{qq}\end{bmatrix}\begin{Bmatrix}\dot{\theta}\\\dot{\boldsymbol{q}}_w\end{Bmatrix} \tag{8}$$

where

$$\boldsymbol{m}_{\theta\theta}(\boldsymbol{q}) = \sum_{p=1}^{n_w} m_p(\bar{\boldsymbol{u}}_{wp})^{\mathrm{T}}\bar{\boldsymbol{I}}_a\bar{\boldsymbol{u}}_{wp} + 2\bar{\bar{\boldsymbol{I}}}_o\boldsymbol{q}_w + (\boldsymbol{q}_w)^{\mathrm{T}}\boldsymbol{m}_{ff}\boldsymbol{q}_w \tag{9a}$$

$$\boldsymbol{m}_{\theta q}(\boldsymbol{q}) = (\boldsymbol{m}_{q\theta}(\boldsymbol{q}))^{\mathrm{T}} = \sum_{p=1}^{n_w} m_p(\bar{\boldsymbol{u}}_{wp})^{\mathrm{T}}\bar{\boldsymbol{I}}_b\boldsymbol{S}_{wp} + (\boldsymbol{q}_w)^{\mathrm{T}}\widetilde{\boldsymbol{S}} \tag{9b}$$

$$\boldsymbol{m}_{qq} = \sum_{p=1}^{n_w} m_p(\boldsymbol{S}_{wp})^{\mathrm{T}}\boldsymbol{S}_{wp} \tag{9c}$$

$$\begin{cases}\bar{\bar{\boldsymbol{I}}}_o = \sum\limits_{p=1}^{n_w} m_p(\bar{\boldsymbol{u}}_{wp})^{\mathrm{T}}\bar{\boldsymbol{I}}_a\boldsymbol{S}_{wp}\\[2mm]\boldsymbol{m}_{ff} = \sum\limits_{p=1}^{n_w} m_p(\boldsymbol{S}_{wp})^{\mathrm{T}}\bar{\boldsymbol{I}}_a\boldsymbol{S}_{wp}\\[2mm]\widetilde{\boldsymbol{S}} = \sum\limits_{p=1}^{n_w} m_p(\boldsymbol{S}_{wp})^{\mathrm{T}}\bar{\boldsymbol{I}}_b\boldsymbol{S}_{wp}\end{cases} \tag{9d}$$

$$\begin{cases}\bar{\boldsymbol{I}}_a = (\boldsymbol{A}_\theta)^{\mathrm{T}}\boldsymbol{A}_\theta = \begin{bmatrix}1 & 0 & 0\\0 & 1 & 0\\0 & 0 & 0\end{bmatrix}\\[6mm]\bar{\boldsymbol{I}}_b = (\boldsymbol{A}_\theta)^{\mathrm{T}}\boldsymbol{A} = \begin{bmatrix}0 & -1 & 0\\1 & 0 & 0\\0 & 0 & 0\end{bmatrix}\end{cases} \tag{9e}$$

$$\boldsymbol{q} = \begin{Bmatrix}\theta\\\boldsymbol{q}_w\end{Bmatrix} \tag{9f}$$

The symbol $m_p$ stands for the lumped mass at the node $p$, and $n_w$ is the number of the active nodes in the FEM. It is noted that $\boldsymbol{M}_w(\boldsymbol{q})$ is a time-varying mass matrix since it is dependent on the time-varying general coordinate vector $\boldsymbol{q}$.

Assume that the rotation angle of the wing is $\theta = 0$ deg at the beginning of the morphing process. The rotational motion of the wing is driven by a preloaded actuator spring whose nominal stiffness coefficient is denoted as $K_{\text{act}}$. As shown in Figure 1b, when reaching $\theta_{\max}$ for the first time, a spring with a very large stiffness coefficient $\alpha_{\text{act}}K_{\text{act}}$ is used to simulate the locking mechanism. The coefficient $\alpha_{\text{act}}$ is a constant greater than 1. Let $t_0$ be the locking time when the rotation angle reaches $\theta_{\max}$, then the variation of the torsional spring constant $K_s$ with the time $t$ can be depicted by Figure 2.

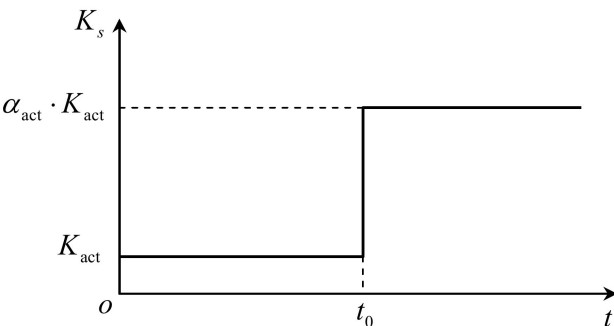

**Figure 2.** Variation of the spring constant with time during the morphing process.

Obviously, the potential energy includes two parts: the elastic potential energy $U_{\text{act}}$ relating to the torsional spring and the strain energy $U_s$ resulting from the elastic deformation of the wing. From Figure 2, we can see that the elastic potential energy $U_{\text{act}}$ is time-dependent and can be written as

$$U_{\text{act}} = \begin{cases} \frac{1}{2}K_{\text{act}}(\theta_{\text{preload}} - \theta)^2, & t < t_0 \\[2mm] \frac{1}{2}\alpha_{\text{act}}K_{\text{act}}(\theta - \theta_{\text{max}})^2, & t \geq t_0 \end{cases} \tag{10}$$

where $\theta_{\text{preload}}$ is the preloaded spring angle.

The strain energy $U_s$ can be written as

$$U_s = \frac{1}{2}(\boldsymbol{q}_w)^{\text{T}}\boldsymbol{K}_{qq}\boldsymbol{q}_w \tag{11}$$

where $\boldsymbol{K}_{qq}$ is the modal stiffness matrix in the floating frame.

Now, the total potential energy of the system can be written as

$$U_w = U_{\text{act}} + U_s = \begin{cases} \frac{1}{2}K_{\text{act}}(\theta_{\text{preload}} - \theta)^2 + \frac{1}{2}(\boldsymbol{q}_w)^{\text{T}}\boldsymbol{K}_{qq}\boldsymbol{q}_w, & t < t_0 \\[2mm] \frac{1}{2}\alpha_{\text{act}}K_{\text{act}}(\theta - \theta_{\text{max}})^2 + \frac{1}{2}(\boldsymbol{q}_w)^{\text{T}}\boldsymbol{K}_{qq}\boldsymbol{q}_w, & t \geq t_0 \end{cases} \tag{12}$$

The dissipation function of the system can be expressed as

$$D_w = D_{\text{act}} + D_s = \begin{cases} \frac{1}{2}C_{\text{act}}\dot{\theta}^2 + \frac{1}{2}(\dot{\boldsymbol{q}}_w)^{\text{T}}\boldsymbol{C}_{qq}\dot{\boldsymbol{q}}_w, & t < t_0 \\[2mm] \frac{1}{2}C_{\text{lock}}\dot{\theta}^2 + \frac{1}{2}(\dot{\boldsymbol{q}}_w)^{\text{T}}\boldsymbol{C}_{qq}\dot{\boldsymbol{q}}_w, & t \geq t_0 \end{cases} \tag{13}$$

where $C_{\text{act}}$ and $C_{\text{lock}}$ are the damping coefficients of the rotational degrees of freedom of the wing in the rotation and the locking phases, respectively. $\boldsymbol{C}_{qq}$ is the modal damping matrix corresponding to the elastic vibration of the wing.

The Lagrange's equation takes the form

$$\frac{\text{d}}{\text{d}t}\left(\frac{\partial T_w}{\partial \dot{\boldsymbol{q}}}\right) - \frac{\partial T_w}{\partial \boldsymbol{q}} + \frac{\partial D_w}{\partial \dot{\boldsymbol{q}}} + \frac{\partial U_w}{\partial \boldsymbol{q}} = \boldsymbol{F}_{\text{gen}} \tag{14}$$

Substituting Equations (8), (12) and (13) into Equation (14), the nonlinear and time-varying equations of the motion for the rotating wing can be written as

$$\boldsymbol{M}_w(\boldsymbol{q})\ddot{\boldsymbol{q}}(t) + \boldsymbol{D}_w\dot{\boldsymbol{q}}(t) + \boldsymbol{K}_w\boldsymbol{q}(t) = \boldsymbol{F}_{\text{const}} + \boldsymbol{F}_{sv}(\boldsymbol{q}, \dot{\boldsymbol{q}}) + \boldsymbol{F}_a \tag{15}$$

where

$$
\begin{cases}
M_w(q) = \begin{bmatrix} m_{\theta\theta} & m_{\theta q} \\ m_{q\theta} & m_{qq} \end{bmatrix} \\[12pt]
D_w = \begin{bmatrix} C_{as} & 0 \\ 0 & C_{qq} \end{bmatrix} \\[12pt]
K_w = \begin{bmatrix} K_{as} & 0 \\ 0 & K_{qq} \end{bmatrix}
\end{cases}
\tag{16}
$$

$$
\begin{cases}
C_{as} = \begin{cases} C_{\text{act}}, & t < t_0 \\ C_{\text{lock}}, & t \geq t_0 \end{cases} \\[12pt]
K_{as} = \begin{cases} K_{\text{act}}, & t < t_0 \\ \alpha_{\text{act}} K_{\text{act}}, & t \geq t_0 \end{cases}
\end{cases}
\tag{17}
$$

The vector $F_{\text{const}}$ in Equation (15) refers to the constant moment related to the preloaded and the locking springs, given by

$$
\begin{cases}
F_{\text{const}} = \begin{Bmatrix} M_{\text{const}} \\ 0 \end{Bmatrix} \\[12pt]
M_{\text{const}} = \begin{cases} K_{\text{act}} \theta_{\text{preload}}, & t < t_0 \\ \alpha_{\text{act}} K_{\text{act}} \theta_{\text{max}}, & t \geq t_0 \end{cases}
\end{cases}
\tag{18}
$$

The vector $F_{sv}(q, \dot{q})$ in Equation (15) can be written as

$$
F_{sv}(q, \dot{q}) = -\dot{M}_w(q)\dot{q} + \frac{\partial}{\partial q}\left(\frac{1}{2}\dot{q}^{\mathrm{T}} M_w(q)\dot{q}\right)
\tag{19}
$$

where $F_{sv}(q, \dot{q})$ is a quadratic velocity vector resulting from the differentiation of the kinetic energy with respect to the time and with respect to the body coordinates. This quadratic velocity vector contains the gyroscopic and Coriolis force components. In the present planar analysis, the components of the vector $F_{sv}(q, \dot{q})$ can be written as

$$
F_{sv}(q, \dot{q}) = \begin{Bmatrix} (F_{sv})_\theta \\ (F_{sv})_{q_w} \end{Bmatrix} = \begin{Bmatrix} -2\dot{\theta}(\dot{q}_w)^{\mathrm{T}}((\bar{\bar{I}}_o)^{\mathrm{T}} + m_{ff}q_w) \\ \dot{\theta}^2((\bar{\bar{I}}_o)^{\mathrm{T}} + m_{ff}q_w) + 2\dot{\theta}\widetilde{S}\dot{q}_w \end{Bmatrix}
\tag{20}
$$

During the derivation of Equation (20), the relation $(\dot{q}_w)^{\mathrm{T}}\widetilde{S}\dot{q}_w = 0$ is used, because $\widetilde{S}$ is a skew symmetric matrix. Note that the quadratic velocity vector that includes the effect of the Coriolis and centrifugal forces is a nonlinear function of the generalized coordinates and velocities. $F_a$ in Equation (15) is the vector of the generalized unsteady aerodynamic forces (GAF) that will be discussed in detail in the next section.

### 2.3. Generalized Unsteady Aerodynamic Forces

In the global frame $oxyz$, the aerodynamic vector equivalent to the node $p$ is represented by a $3 \times 1$ vector $f_{ap}$. From Equation (1), the virtual displacement of the node $p$ can be written as

$$
\delta r_p = [A_\theta(\theta)(\bar{u}_{wp} + S_{wp}q_w(t)) \quad A(\theta)S_{wp}]\begin{Bmatrix} \delta\theta \\ \delta q_w \end{Bmatrix}
\tag{21}
$$

The total virtual work performed by the aerodynamic forces acting on the finite element nodes can be expressed as

$$
\begin{aligned}
\delta W &= \sum_{p=1}^{n_s} (f_{ap})^{\mathrm{T}} \delta r_p \\
&= \sum_{p=1}^{n_s} (f_{ap})^{\mathrm{T}} [A_\theta(\theta)(\bar{u}_{wp} + S_{wp} q_w(t)) \quad A(\theta) S_{wp}] \left\{ \begin{matrix} \delta\theta \\ \delta q_w \end{matrix} \right\} \\
&= \begin{bmatrix} (Q_a)_\theta \\ (Q_a)_{q_w} \end{bmatrix}^{\mathrm{T}} \left\{ \begin{matrix} \delta\theta \\ \delta q_w \end{matrix} \right\} = F_a^{\mathrm{T}} \left\{ \begin{matrix} \delta\theta \\ \delta q_w \end{matrix} \right\}
\end{aligned}
\tag{22}
$$

where

$$
F_a = \begin{bmatrix} (Q_a)_\theta \\ (Q_a)_{q_w} \end{bmatrix}
\tag{23}
$$

$$
(Q_a)_\theta = \sum_{p=1}^{n_s} (f_{ap})^{\mathrm{T}} A_\theta(\theta)(\bar{u}_{wp} + S_{wp} q_w(t))
\tag{24}
$$

$$
(Q_a)_{q_w} = \left( \sum_{p=1}^{n_s} (f_{ap})^{\mathrm{T}} A(\theta) S_{wp} \right)^{\mathrm{T}} = \sum_{p=1}^{n_s} (S_{wp})^{\mathrm{T}} A^{\mathrm{T}}(\theta)(f_{ap})
\tag{25}
$$

The symbol $n_s$ in the summation represents the number of the user-selected finite element nodes on which the equivalent aerodynamic forces are calculated.

Note that the position vector $\bar{u}_{wp}$ of the finite element nodes, the vibration mode data $S_{wp}$, and the transformation matrices $A$ and $A_\theta$ can be obtained in advance. It can be seen from Equations (24) and (25) that once the aerodynamic force $f_{ap}$ and the generalized force $(Q_a)_\theta$ are obtained, $(Q_a)_{q_w}$ can then be calculated.

The complexity of the aeroelastic problem lies in that the aerodynamic force vector $f_{ap}$ is dependent on the motion of the structure. Hence, the interpolation between the aerodynamic force and the structural motion is required.

In this paper, the local piston theory is used to calculate the unsteady aerodynamic forces of a variable swept wing [31]. At a high Mach number, the piston theory assumes that the disturbance of an airfoil and flow field is similar to the piston motion. Due to the high Mach number effect, the spatial characteristic of the flow shows a strong local effect, so that the pressure on the airfoil and the downwash boundary conditions at the point forms a one-to-one mapping. Furthermore, the temporal characteristic of the flow shows a weak memory effect. These features make the expressions of the piston theory very simple and concise.

As shown in Figure 3, in the global frame $oxyz$, the nodal force vector $f_{ap}$ can be written as

$$
f_{ap} = \left\{ \begin{matrix} f_{dp} \cos\alpha \\ 0 \\ f_{lp} + f_{dp} \sin\alpha \end{matrix} \right\} = f_{lp} \left\{ \begin{matrix} 0 \\ 0 \\ 1 \end{matrix} \right\} + f_{dp} \left\{ \begin{matrix} \cos\alpha \\ 0 \\ \sin\alpha \end{matrix} \right\}
\tag{26}
$$

where $f_{lp}$ and $f_{dp}$ are the lift and drag acting on the node $p$, respectively. It can be seen that Equation (26) is divided into two parts: one is related to the normal force of the node and the other is related to the drag.

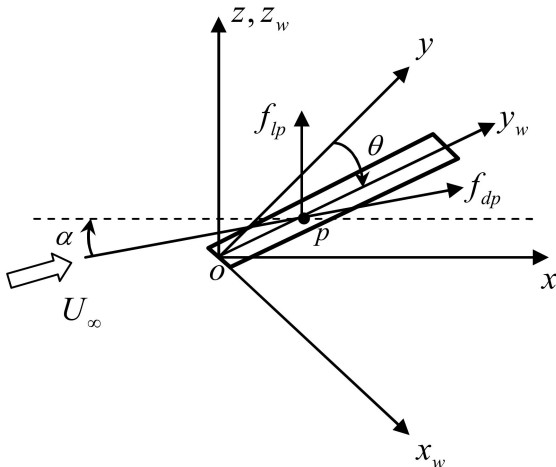

**Figure 3.** Aerodynamic forces equivalent to the node $p$.

To calculate $(Q_a)_\theta$ given by Equation (24), substitute Equation (26) into Equation (24), and we obtain

$$
(Q_a)_\theta =
$$

$$
\sum_{p=1}^{n_{sl}} f_{lp}
\begin{bmatrix} 0 \\ 0 \\ 1 \end{bmatrix}^{\mathrm{T}}
\begin{bmatrix} -\sin\theta & \cos\theta & 0 \\ -\cos\theta & -\sin\theta & 0 \\ 0 & 0 & 0 \end{bmatrix}
(\bar{u}_{wp} + S_{wp} q_w(t))
$$

$$
+ \sum_{p=1}^{n_{sd}} f_{dp}
\begin{bmatrix} \cos\alpha \\ 0 \\ \sin\alpha \end{bmatrix}^{\mathrm{T}}
\begin{bmatrix} -\sin\theta & \cos\theta & 0 \\ -\cos\theta & -\sin\theta & 0 \\ 0 & 0 & 0 \end{bmatrix}
(\bar{u}_{wp} + S_{wp} q_w(t))
\tag{27}
$$

where $n_{sl}$ represents the number of the finite element nodes used for the equivalence of the aerodynamic forces. $n_{sd}$ represents the number of the finite element nodes used for the equivalence of the aerodynamic drag.

Note that the first summation term in Equation (27) is zero, and Equation (27) can be rewritten as

$$
(Q_a)_\theta = \sum_{p=1}^{n_{sd}} f_{dp}
\begin{bmatrix} -\sin\theta\cos\alpha \\ \cos\theta\cos\alpha \\ 0 \end{bmatrix}^{\mathrm{T}}
(\bar{u}_{wp} + S_{wp} q_w(t))
\tag{28}
$$

In general, the distribution of drag can be written as

$$
\frac{\mathrm{d}D}{\mathrm{d}r} = \frac{1}{2}\rho_\infty U_\infty^2 c_d f_{cr}(\theta)
\tag{29}
$$

where $\mathrm{d}r$ is the infinitesimal length in the span, and $\mathrm{d}D$ is the drag forces acting on the strip with the infinitesimal length $\mathrm{d}r$. $f_{cr}(\theta)$ is the correction factor, which is the function of the sweep angle $\theta$. $c_d$ is the drag coefficient, given by

$$
c_d = c_{d0} + c_{d2}\alpha^2 + c_{d4}\alpha^4
\tag{30}
$$

in which $c_{d0}$, $c_{d2}$ and $c_{d4}$ are constants.

To calculate $(Q_a)_{q_w}$ given by Equation (25), substitute Equation (26) into Equation (25) to obtain

$$(Q_a)_{q_w} = \left( \sum_{p=1}^{n_s} \left( f_{ap} \right)^{\mathrm{T}} A(\theta) S_{wp} \right)^{\mathrm{T}} = \sum_{p=1}^{n_s} \left( S_{wp} \right)^{\mathrm{T}} A^{\mathrm{T}}(\theta) \left( f_{ap} \right)$$

$$= \sum_{p=1}^{n_{sl}} f_{lp} \left( S_{wp} \right)^{\mathrm{T}} A^{\mathrm{T}}(\theta) \begin{bmatrix} 0 \\ 0 \\ 1 \end{bmatrix} + \sum_{p=1}^{n_{sd}} f_{dp} \left( S_{wp} \right)^{\mathrm{T}} A^{\mathrm{T}}(\theta) \begin{bmatrix} \cos \alpha \\ 0 \\ \sin \alpha \end{bmatrix} \tag{31}$$

$$= \sum_{p=1}^{n_{sl}} \left( S_{wp} \right)^{\mathrm{T}} \begin{bmatrix} 0 \\ 0 \\ f_{lp} \end{bmatrix} + \sum_{p=1}^{n_{sd}} f_{dp} \left( S_{wp} \right)^{\mathrm{T}} \begin{bmatrix} \cos \theta \cos \alpha \\ \sin \theta \cos \alpha \\ \sin \alpha \end{bmatrix}$$

$$= Q_{al} + Q_{ad}$$

Note that the second summation term $Q_{ad}$ in Equation (31) is easy to calculate, so the following discussion will focus on the calculation of the first summation term $Q_{al}$, which represents the modal aerodynamic forces in the $z$ direction.

The unsteady aerodynamic forces are calculated based on the Van Dyke second-order piston theory. In the global frame $oxyz$, the local differential pressure coefficient $\Delta c_p(x, y, t)$ of the lifting surface can be written as [25]

$$\Delta c_p(x, y, t) = -\frac{4}{M_\infty} A_a(x, y) \left( -\alpha + \frac{\partial Z(x, y, t)}{\partial x} + \frac{1}{U_\infty} \frac{\partial Z(x, y, t)}{\partial t} \right) \tag{32}$$

where

$$A_a(x, y) = c_1 + 2 c_2 M_\infty \frac{\partial H(x, y)}{\partial x} \tag{33}$$

$$\begin{cases} c_1 = \frac{M_\infty}{\beta} \\ c_2 = \frac{M_\infty^4 (\gamma + 1) - 4\beta^2}{4\beta^4} \\ \beta = \sqrt{M_\infty^2 - 1} \, . \end{cases} \tag{34}$$

The function $H(x, y)$ represents the airfoil thickness, $M_\infty$ is the Mach number, $Z(x, y, t)$ is the displacement of a node in the surface of the wing in the $z_w$ direction, and $\alpha$ is the static angle of the attack.

In order to account for the influence of the time-varying sweep angle, the coefficients $c_1$ and $c_2$ in Equation (33) can take the following form [26]:

$$\begin{cases} c_1 = \frac{M_\infty}{\sqrt{M_\infty^2 - \sec^2 \Lambda}}, \\ c_2 = \frac{[M_\infty^4 (\gamma + 1) - 4 \sec^2 \Lambda \cdot (M_\infty^2 - \sec^2 \Lambda)]}{[4(M_\infty^2 - \sec^2 \Lambda)^2]} . \end{cases} \tag{35}$$

where $\Lambda$ is the leading edge sweep angle.

It is noted that the calculation of the downstream slope $\partial Z(x, y, t) / \partial x$ in Equation (32) requires the mode shape data of the wing under an arbitrary rotation angle $\theta$, which is very inconvenient for the numerical simulation. To improve the computational efficiency, the principle of relative motion can be used. As shown in Figure 4, the aerodynamic force acting on the rotating wing can be calculated by continuously changing the deflection angle $\theta$ of the air flow according to the structural and aerodynamic models in the local rotating coordinate system. In this way, the normal unsteady aerodynamic forces of the rotating wing can be calculated in the $ox_w y_w z_w$ frame. Meanwhile, the FEM of the wing structure needs to be established only once in the frame. The natural vibration characteristics of the wing can be obtained from the FEM in the local floating frame. Assuming that the first $n$

order natural modes of the structure are retained, the displacement of the wing in the $z_w$ direction can be expressed as

$$Z(x_w, y_w, t) = \sum_{j=1}^{n} f_j(x_w, y_w) q_{wj}(t) \tag{36}$$

where $f_j(x_w, y_w)$ is the $j$-th mode shape of the wing in the $z_w$ direction, and $q_{wj}(t)$ is the modal coordinate.

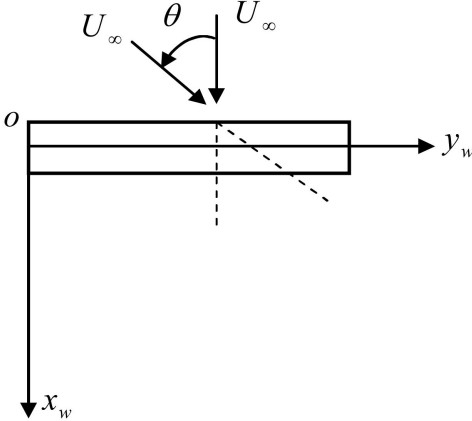

**Figure 4.** Rotation of the air flow.

In the presence of the deflection angle of the air flow, the local pressure difference at a point $(x_w, y_w)$ of the wing is

$$\Delta p(\theta, x_w, y_w, t) = -\frac{4q_d}{M_\infty} \sum_{j=1}^{n} \left[ A_a(\theta, x_w, y_w)(-\alpha + \frac{\partial f_j(x_w, y_w)}{\partial \xi} q_{wj}(t) + \frac{1}{U_\infty} f_j(x_w, y_w) \dot{q}_{wj}(t)) \right] \tag{37}$$

where

$$A_a(\theta, x_w, y_w) = c_1 + 2c_2 M_\infty \frac{\partial H(x_w, y_w)}{\partial \xi} \tag{38}$$

$c_1$ and $c_2$ are given by Equation (35), in which $\Lambda = \theta$.

According to Figure 5, there exists the following transformations between the coordinate system $o\xi\eta$ that indicate the direction of air flow and the local coordinate system $ox_w y_w$:

$$x_w = \xi \cos \theta - \eta \sin \theta, \ y_w = \eta \cos \theta + \xi \sin \theta \tag{39}$$

We have

$$
\begin{aligned}
\frac{\partial f_j(x_w, y_w)}{\partial \xi} &= \frac{\partial f_j(x_w, y_w)}{\partial x_w} \frac{\partial x_w}{\partial \xi} + \frac{\partial f_j(x_w, y_w)}{\partial y_w} \frac{\partial y_w}{\partial \xi} \\
&= \frac{\partial f_j(x_w, y_w)}{\partial x_w} \cos \theta + \frac{\partial f_j(x_w, y_w)}{\partial y_w} \sin \theta
\end{aligned} \tag{40}
$$

Substitute Equation (40) into Equation (37), the local pressure difference at a point $(x_w, y_w)$ of the wing under the air flow deflection angle $\theta$ can be written as

$$\Delta p(\theta, x_w, y_w, t)$$

$$= -\frac{4q_d}{M_\infty} \sum_{j=1}^{n} \left[ (-\alpha + (\frac{\partial f_j(x_w, y_w)}{\partial x_w} \cos \theta + \frac{\partial f_j(x_w, y_w)}{\partial y_w} \sin \theta) q_{wj}(t) + \frac{1}{U_\infty} f_j(x_w, y_w) \dot{q}_{wj}(t)) \right] \tag{41}$$

where

$$A_a(\theta, x_w, y_w) = c_1 + 2c_2 M_\infty \left( \frac{\partial H(x_w, y_w)}{\partial x_w} \cos\theta + \frac{\partial H(x_w, y_w)}{\partial y_w} \sin\theta \right) \tag{42}$$

Therefore, the *i*-th modal aerodynamic force $Q_{ai}$ can be written as

$$Q_{ai} = \iint \Delta p(\theta, x_w, y_w, t) f_i(x_w, y_w) \mathrm{d}x_w \mathrm{d}y_w = -\frac{4q_d}{M_\infty} \sum_{j=1}^{n} \left[ A_{ij}(\theta) q_{wj}(t) + \frac{1}{U_\infty} B_{ij}(\theta) \dot{q}_{wj}(t) \right] \tag{43}$$

where

$$A_{ij}(\theta) = \iint A_a(\theta, x_w, y_w)(-\alpha + g_j(\theta, x_w, y_w)) f_i(x_w, y_w) \mathrm{d}x_w \mathrm{d}y_w \tag{44a}$$

$$B_{ij}(\theta) = \iint A_a(\theta, x_w, y_w) f_j(x_w, y_w) f_i(x_w, y_w) \mathrm{d}x_w \mathrm{d}y_w \tag{44b}$$

$$g_j(\theta, x_w, y_w) = \frac{\partial f_j(x_w, y_w)}{\partial x_w} + \frac{\partial f_j(x_w, y_w)}{\partial y_w} \tag{44c}$$

It can be seen from Equation (44a–c) that $A_{ij}$ and $B_{ij}$ depend on the mode shape, the thickness function of the lifting surface and the rotation angle $\theta$.

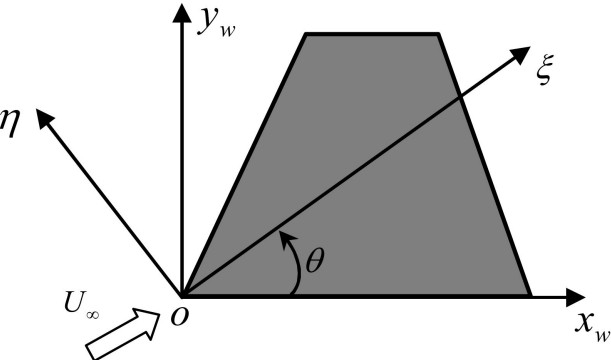

**Figure 5.** Definition of the deflection angle $\theta$ of the air flow.

Now, the modal aerodynamic force vector $\boldsymbol{Q}_{al}$ in Equation (31) can be written in the following form:

$$\boldsymbol{Q}_{al} = \begin{bmatrix} Q_{a1} \\ Q_{a2} \\ \vdots \\ Q_{ai} \\ \vdots \\ Q_{an} \end{bmatrix} = q_d \left( \overline{\boldsymbol{A}}(\theta) \boldsymbol{q}_w(t) + \frac{1}{U_\infty} \overline{\boldsymbol{B}}(\theta) \dot{\boldsymbol{q}}_w(t) \right) \tag{45}$$

where the elements in matrices $\overline{\boldsymbol{A}}(\theta)$ and $\overline{\boldsymbol{B}}(\theta)$ are

$$\overline{A}_{ij} = -\frac{4}{M_\infty} A_{ij}(\theta), \ \overline{B}_{ij} = -\frac{4}{M_\infty} B_{ij}(\theta) \tag{46}$$

*2.4. Nonlinear and Time-Varying Aeroelastic Equations of the Variable Swept Wing*

Note that the unsteady aerodynamic vector $F_a$ can be written as the following form:

$$F_a = \begin{bmatrix} (Q_a)_\theta \\ (Q_a)_{q_w} \end{bmatrix} = \begin{bmatrix} 0 \\ Q_{al} \end{bmatrix} + \begin{bmatrix} (Q_a)_\theta \\ Q_{ad} \end{bmatrix}$$

$$= F_{al}(q, \dot{q}) + F_{ad}(q) \tag{47}$$

where

$$F_{al}(q, \dot{q}) = \begin{bmatrix} 0 \\ Q_{al} \end{bmatrix}, \; F_{ad}(q) = \begin{bmatrix} (Q_a)_\theta \\ Q_{ad} \end{bmatrix} \tag{48}$$

According to Equation (45), $F_{al}(q, \dot{q})$ can be written as

$$F_{al}(q, \dot{q}) = \begin{bmatrix} 0 \\ Q_{al} \end{bmatrix} = \frac{1}{2}\rho_\infty U_\infty^2 \overline{\overline{A}}(\theta)q(t) + \frac{1}{2}\rho_\infty U_\infty \overline{\overline{B}}(\theta)\dot{q}(t) \tag{49}$$

where

$$\overline{\overline{A}}(\theta) = \begin{bmatrix} 0 & 0 \\ 0 & \overline{A}(\theta) \end{bmatrix}, \; \overline{\overline{B}}(\theta) = \begin{bmatrix} 0 & 0 \\ 0 & \overline{B}(\theta) \end{bmatrix} \tag{50}$$

Based on Equations (15), (18), (19) and (47), the nonlinear and time-varying aeroelastic equations of the variable-swept wing can be obtained as

$$M_w(q)\ddot{q}(t) + (D_w - \tfrac{1}{2}\rho_\infty U_\infty \overline{\overline{B}}(\theta))\dot{q}(t) + (K_w - \tfrac{1}{2}\rho_\infty U_\infty^2 \overline{\overline{A}}(\theta))q(t)$$

$$= F_{\text{const}} + F_{sv}(q, \dot{q}) + F_{ad}(q) \tag{51}$$

or in the state space form

$$\dot{\widetilde{X}}(t) = \widetilde{A}(q)\widetilde{X}(t) + \widetilde{F}_{sv}(q, \dot{q}) + \widetilde{F}_{ad}(q) + \widetilde{F}_{\text{const}}(q) \tag{52}$$

where

$$\hat{A}(q) = \begin{bmatrix} 0 & I \\ -M_w^{-1}(q)\left(K_w - \tfrac{1}{2}\rho_\infty U_\infty^2 \overline{\overline{A}}(\theta)\right) & -M_w^{-1}(q)\left(D_w - \tfrac{1}{2}\rho_\infty U_\infty \overline{\overline{B}}(\theta)\right) \end{bmatrix} \tag{53}$$

$$\widetilde{X}(t) = \begin{bmatrix} q(t) \\ \dot{q}(t) \end{bmatrix} \tag{54}$$

$$\begin{cases} \widetilde{F}_{sv} = \begin{bmatrix} 0 \\ M_w^{-1}(q)F_{sv}(q, \dot{q}) \end{bmatrix} \\[2mm] \widetilde{F}_{ad}(q) = \begin{bmatrix} 0 \\ M_w^{-1}(q)F_{ad}(q) \end{bmatrix} \\[2mm] \widetilde{F}_{\text{const}} = \begin{bmatrix} 0 \\ M_w^{-1}(q)F_{\text{const}} \end{bmatrix} \end{cases} \tag{55}$$

*2.5. Several Issues on Numerical Simulations*

2.5.1. Double Numerical Integrations

Note that when calculating matrices $\overline{A}(\theta)$ and $\overline{B}(\theta)$ in Equation (45), it is necessary to calculate the double integrals given by Equation (44a,b), in which the analytical modal shape $f_j(x_w, y_w)$ and the downstream slope should be known in advance. In actual simulations, numerical integration is used for Equation (44a,b). Hence, $f_j(x_w, y_w)$ should be replaced by the discrete modal data obtained by the FEM. In order to deal with the double numerical integrations, the lifting surface of the wing is divided into a number of quadrilateral elements. Based on the isoparametric transformation, the integration problem in the non-rectangular region is transformed into the one in the square region. Afterwards, the Gauss quadrature formula is used to calculate the element integral $I_k$. Finally, the double integral over the entire lifting surface can be expressed as

$$I = \sum_{k=1}^{m} I_k \tag{56}$$

where $m$ is the number of the elements on the lifting surface. It can be seen that the computation of the double integrals only requires values of the integrand at the Gaussian integration points.

Due to the inconsistency between the structure and the aerodynamic grid, it is necessary to transfer the vibration displacement and aerodynamic data between the structure and the aerodynamic grid. To this end, let the spline matrix $G_{as}$ be the transformation matrix from the global FEM displacement vector $u$ to the displacement vector $h_a$ at the interpolation points, so we have the following relationship:

$$h_a = G_{as}u \tag{57}$$

It can be seen from Equation (57) that the functional values at the Gaussian integral points can be obtained by the spline interpolation of the mode shape values at the pre-selected structural finite element nodes.

Similarly, the slope vector $h_\alpha$ at the interpolation points can be written as

$$h_\alpha = G_{\alpha s}u \tag{58}$$

where $G_{\alpha s}$ is the spline matrix.

Obviously, the key to obtain the displacement and slope vectors at the interpolation points is to find the spline matrices $G_{as}$ and $G_{\alpha s}$, which can be formed by the infinite plate spline (IPS) interpolation method [32].

2.5.2. Time-Varying Lifting Surface

As shown in Figure 6, the region that overlaps with the fuselage does not produce lift. Therefore, the wing outside the fuselage is the effective lifting surface that generates the aerodynamic forces. Moreover, the area of the effective lifting surface is time-varying during the morphing process. In Figure 6, for example, when $\theta = 0$ deg, the effective lifting surface is enclosed by $A_0CDB_0$. At an arbitrary rotation angle $\theta$, the effective lifting surface is enclosed by $A_\theta CDB_\theta$.

Therefore, during the rotation of the wing, it is necessary to determine the positions of the left boundary points $A_\theta$ and $B_\theta$ of the effective lifting surface in the floating frame $ox_w y_w$ in real time. The $y_w$ coordinate values of the boundary points $A_\theta$ and $B_\theta$ can be calculated by

$$\begin{cases} y_{wA_\theta} = \frac{l_b - h_a \sin\theta}{\cos\theta} \\ y_{wB_\theta} = \frac{l_b + h_b \sin\theta}{\cos\theta} \end{cases} \tag{59}$$

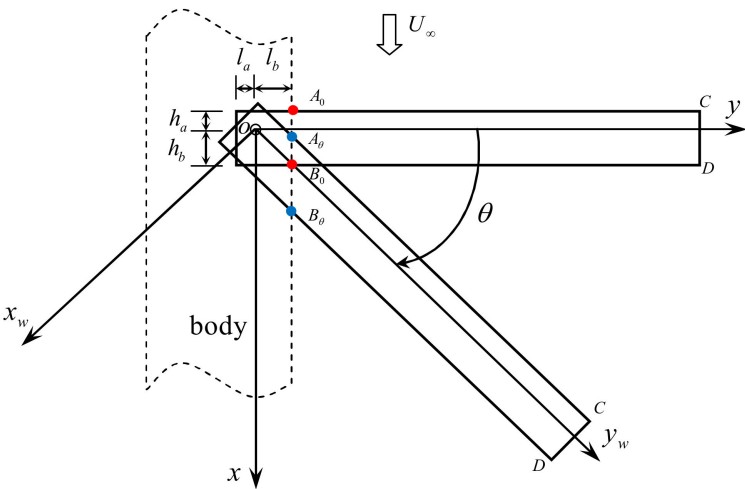

**Figure 6.** The time-varying effective lifting surface.

To demonstrate the concept of the time-varying effective lifting surface of a variable swept wing during the morphing process, consider a rotating rectangle wing with half-span $l_s = 1000$ mm and chord length $c_s = 200$ mm. Assume that the rotation axis of the wing is located at $l_a = 30$ mm, $l_b = 40$ mm, $h_a = 30$ mm and $h_b = 170$ mm. The calculated left boundary points (red circles) of the effective lifting surface are shown in Figure 7. The effective lifting surface (the red region) corresponding to each sweep angle is shown in Figure 8. Obviously, the area of the effective lifting surface is time-varying during the morphing process of the wing.

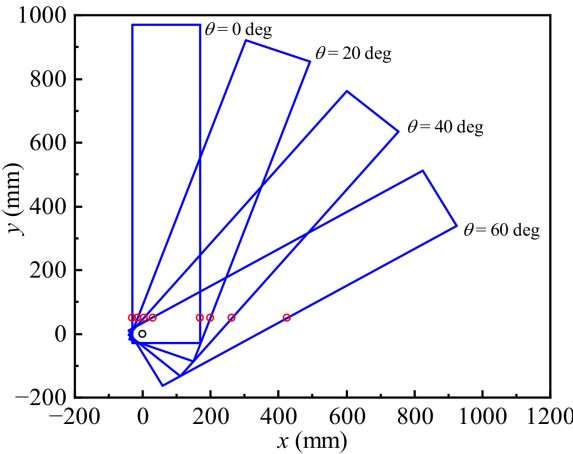

**Figure 7.** The calculated left boundary points (red circles) of the effective lifting surface.

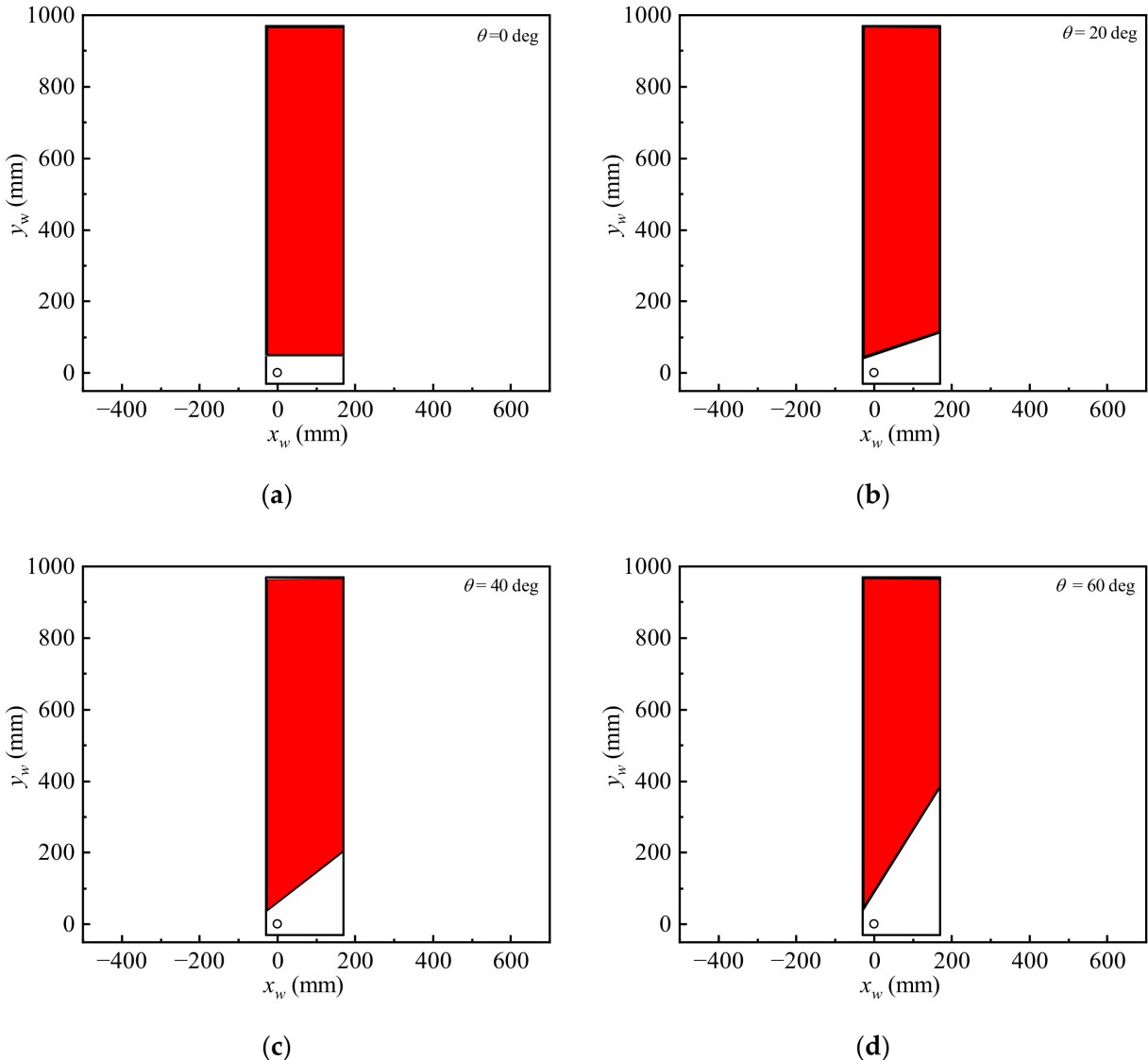

**Figure 8.** The time-varying effective lifting surface. (**a**) Sweep angle is 0 deg. (**b**) Sweep angle is 20 deg. (**c**) Sweep angle is 40 deg. (**d**) Sweep angle is 60 deg.

## 3. Numerical Simulations

This study focuses on the time-varying aeroelastic response behaviors of the missile wing during the rapid rotation at a high flight speed. The considered missile model is shown in Figure 9, in which the wings can be deployed after launching and change the sweep angle according to different flight conditions. The FEM and the aerodynamic models are shown in Figure 9b. The chord length of the wing is 120 mm, the half-span of the wing is 720 mm when it is fully deployed (the configuration at $\theta = 0$ deg), the area of the wing is 0.5957 m$^2$, and the weight is 2.49 kg. The leading edge sweep angle $\theta$ of the wing can vary from 0 deg to a maximum 65 deg.

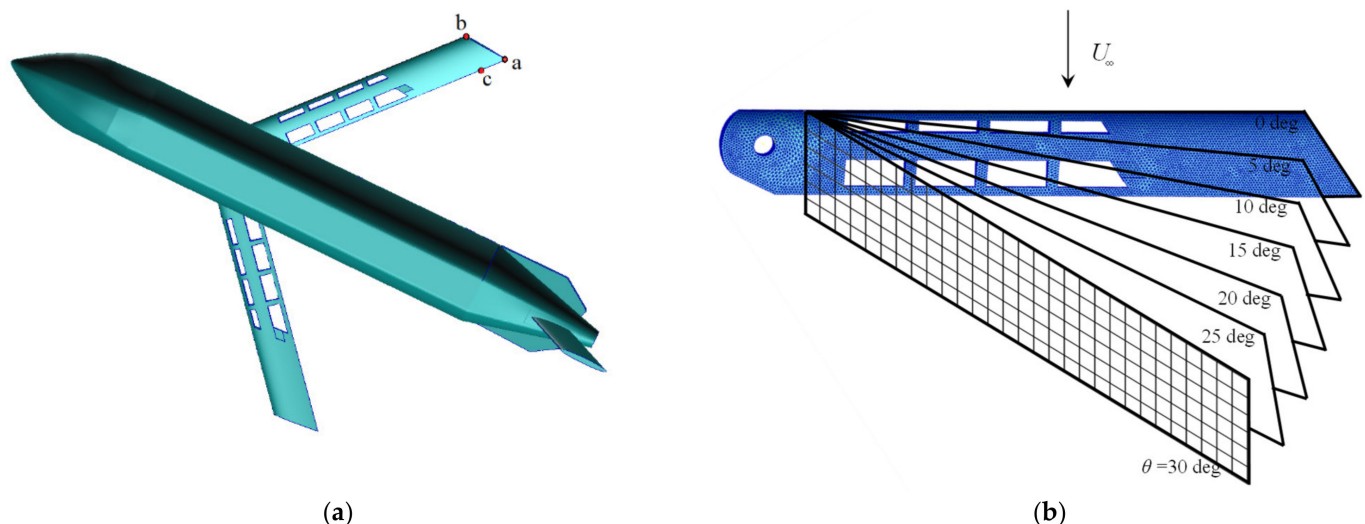

**Figure 9.** The rotating wing models used for simulations. (**a**) The geometric model of the considered missile. (**b**) The FEM and aerodynamic models.

In this study, the FEM of the wing was constructed by MSC.Patran. The node coordinates were obtained from the generated BDF file. The mass, stiffness and mode shape matrices were obtained from the normal mode analysis in MSC.Nastran. These data will be used to construct the developed time-varying aeroelastic model of the rotating wing. Table 1 gives the first ten natural frequencies of the wing. Notably, the fuselage and root of the wing are fixed in the FEM.

**Table 1.** The first ten natural modes ($\theta = 0$ deg).

| Order | Frequency (Hz) | Mode Shape |
|---|---|---|
| Mode 1 | 14.34 | 1st vertical bending |
| Mode 2 | 81.89 | 1st in-plane bending |
| Mode 3 | 117.43 | 2nd vertical bending |
| Mode 4 | 204.65 | 1st torsion |
| Mode 5 | 363.33 | 3rd vertical bending |
| Mode 6 | 544.56 | 2nd torsion |
| Mode 7 | 682.44 | 4th vertical bending |
| Mode 8 | 909.84 | 3rd torsion |
| Mode 9 | 1049.79 | 5th vertical bending |
| Mode 10 | 1245.86 | 4th torsion |

*3.1. Flutter Analysis*

Predicting the flutter stability of the system is the premise of the time-varying aeroelastic simulations. To this end, the flutter velocity and the flutter mode at different sweep angles for a fixed Mach number ($M_\infty = 3.0$) were calculated by using the *p-k* method. It can be seen from Figure 10a that the flutter characteristics of the system are strongly dependent on the sweep angle, which can be attributed to the changes of the effective lift surface and its downstream slope under different sweep angles. With the increase of the sweep angle, the flutter speed of the wing first decreases slightly and then gradually increases until the sweep angle reaches 58 deg, after which the flutter speed decreases rapidly. Additionally, a small jump of the flutter speed occurs between 57 deg and 58 deg. It can be found from the computational results by the software, and the approach used in the paper is also confirmed by comparison in some way. In order to study the effect of the rotational speed on the variable-swept wing, this paper studies the contribution of rotational speeds for the flutter speed. The rotational rate could strength the stiffness of the structure and improve the

flutter speed at high rotational speeds, as you see in Figure 10b. However, it is hard for a variable-swept wing to reach a very high speed (16 rad/s is a reference to a rotational speed of the blade on a normal helicopter). The highest rotational speed in this paper is less than 8 rad/s, the effect of which is less than 1% on the wing (the highest speed occurs near 41 deg). In short, the flutter speed of the system undergoes a large change in the range of the given sweep angle, so special attention should be paid to the aeroelastic stability in the design of the variable-swept wings.

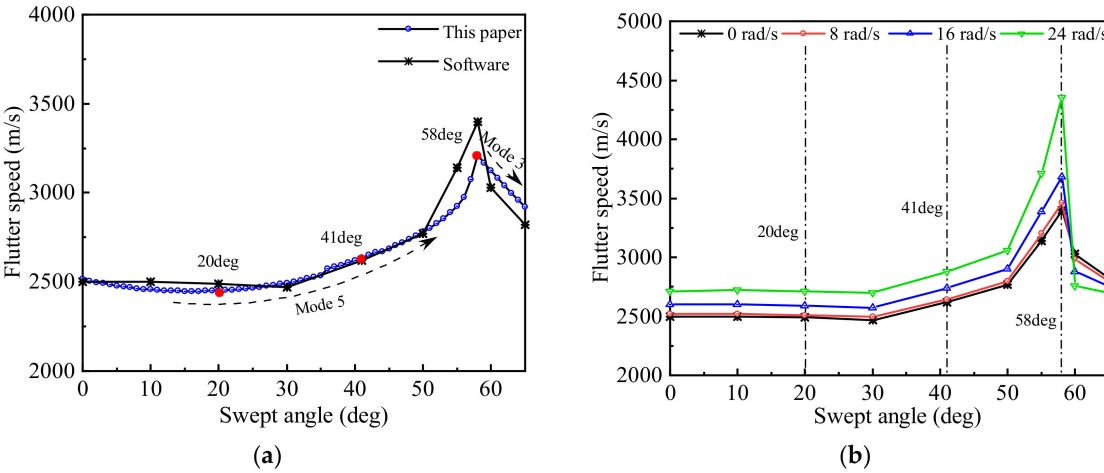

(**a**)                    (**b**)

**Figure 10.** Variations of the flutter speed with the sweep angle. (**a**) This paper vs. software at fixed angle. (**b**) The flutter speed varies with different rotational speeds.

In order to analyze the flutter character, Figures 11 and 12 provide structural damping varying with the incoming flow speeds for the first six modes (1st, 2nd, 3rd, 4th and 6th). When the sweep angle is less than 57 deg, the fifth-order mode becomes unstable. In this range, the flutter is characterized by the hump shape, as shown in Figure 11, in which the unstable mode crosses the zero point (g = 0) twice. Figure 12 shows the flutter curves at the sweep angle of 57 deg. It can be seen that, with the increase of the flow speed, the fifth-order modal branch reaches the zero point, and then drops rapidly, which indicates that a switch of the unstable mode is coming. When the sweep angle increases to 58 deg, the third-order modal branch first crosses the zero point and becomes an unstable mode. Therefore, the essential reason for the jumping phenomenon of flutter speed is the switching of the unstable modal branches.

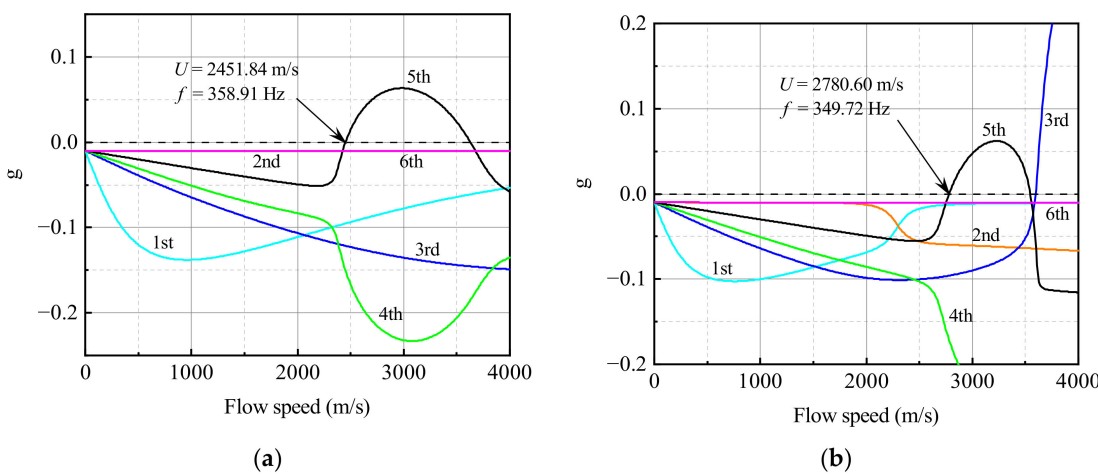

(**a**)                    (**b**)

**Figure 11.** Flutter characteristics at different sweep angles. (**a**) Sweep angle is 20 deg. (**b**) Sweep angle is 50 deg.

Note that under the high sweep angle, the unstable modal branch has a steep slope at the zero point, as shown in Figure 12. This type of flutter is more dangerous compared with the hump flutter, and can be referred to as the explosive flutter.

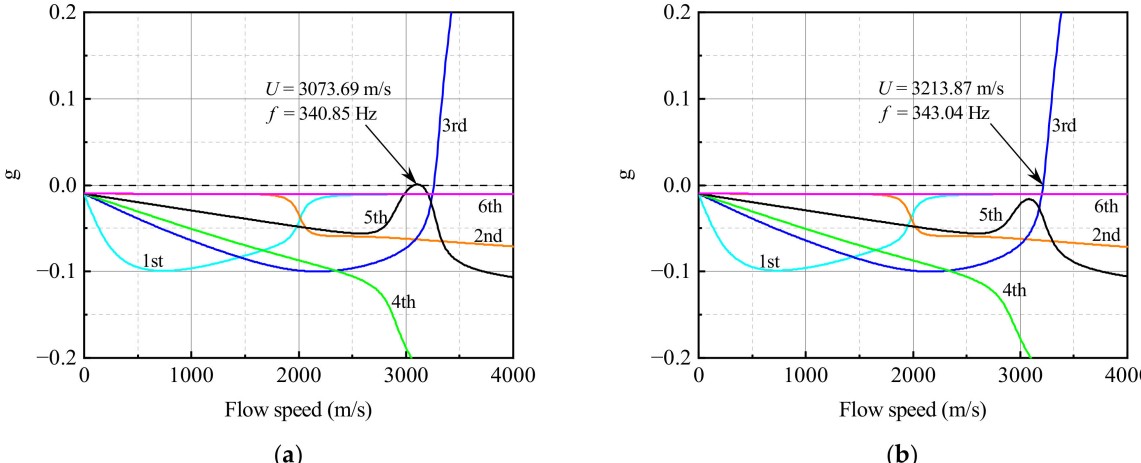

**Figure 12.** Change of the unstable mode under different sweep angles. (**a**) Sweep angle is 57 deg. (**b**) Sweep angle is 58 deg.

### 3.2. Time-Varying Aeroelastic Responses

Numerical simulations for time-varying aeroelastic responses of the variable-swept wing during the rapid morphing process are carried out at a flight speed lower than the flutter point of the system.

Unless specifically stated, the parameters used in this study are as follows. The flow conditions are sea level altitude and a free stream Mach number of $M_\infty = 3.0$ ($U_\infty = 1020$ m/s). All the simulations are performed at the trim status of the system in which the sweep angle of the wing is $\theta = 0.0$ deg. The relative rotation between the wing and the body (fuselage) about the $z$ axis is implemented by an actuator spring with a pre-torsion angle $\theta_{\text{preload}} = 40$ deg. The rotating motion starts with a sweep angle of 0.0 deg and ends at the locked position $\theta_{\text{max}} = 60$ deg. The stiffness of the preloaded spring is denoted as $K_{\text{act}}$. When reaching the maximum rotation angle $\theta_{\text{max}}$, the actuator spring is replaced by a locking spring with a very large stiffness coefficient $\alpha_{\text{act}} K_{\text{act}}$. The damping ratio of the rotation motion can be varied with different operational phases of the wing, given by

$$\begin{cases} \zeta_{\text{act}} = \dfrac{C_{\text{act}}}{2\sqrt{I_w \cdot K_{\text{act}}}} \\[3mm] \zeta_{\text{lock}} = \dfrac{C_{\text{lock}}}{2\sqrt{I_w \cdot \alpha_{\text{act}} \cdot K_{\text{act}}}} \end{cases} \tag{60}$$

where $\zeta_{\text{act}}$ and $\zeta_{\text{lock}}$ are the damping ratios during the rotating and the locking phases, respectively. $I_w$ is the moment of inertia of the wing about the z-axis. In simulations, we set $\zeta_{\text{act}} = 0.05$, and $\zeta_{\text{lock}} = 0.05 \sim 0.2$. The first ten clamped natural modes are used to account for the elastic vibrations of the wing during the morphing process. The rotating motion of the wing starts at $t = 0.05$ s.

Figure 13 shows the time histories of the sweep angle and the angular acceleration under different actuator stiffness $K_{act}$ and the same locking spring stiffness. The damping ratio in the licking phase is taken as $\zeta_{lock} = 0.05$. As can be seen, under the driving of the pre-torsional spring, the wing reaches the locking position in a very short time, and then oscillates near this position under the action of the locking spring. The greater the actuator stiffness, the shorter the time to reach the locking position. Additionally, we can see that at the initial stage of wing rotation, there is a fluctuation in the angular acceleration due to the action of the actuator spring, and then it disappears quickly until the wing enters the post-lock state. The angular acceleration gradually decreases in the rotating stage due to the decreasing moment produced by the actuator spring. Figure 14 shows the effect of the damping ratio in the rotational degree of freedom at the post-lock phase on the transient responses of the wing. In simulations, the actuator stiffness is $K_{act} = 20 \, \text{N} \cdot \text{m/deg}$, and the proportional constant is $\alpha_{act} = 100$. We can see that the oscillations decay exponentially due to the damping effect in the post-lock state. The greater the damping ratio is, the faster the vibration attenuations. Hence, the post-lock vibrations can be suppressed through the proper design of the damping level in the rotational degree of freedom.

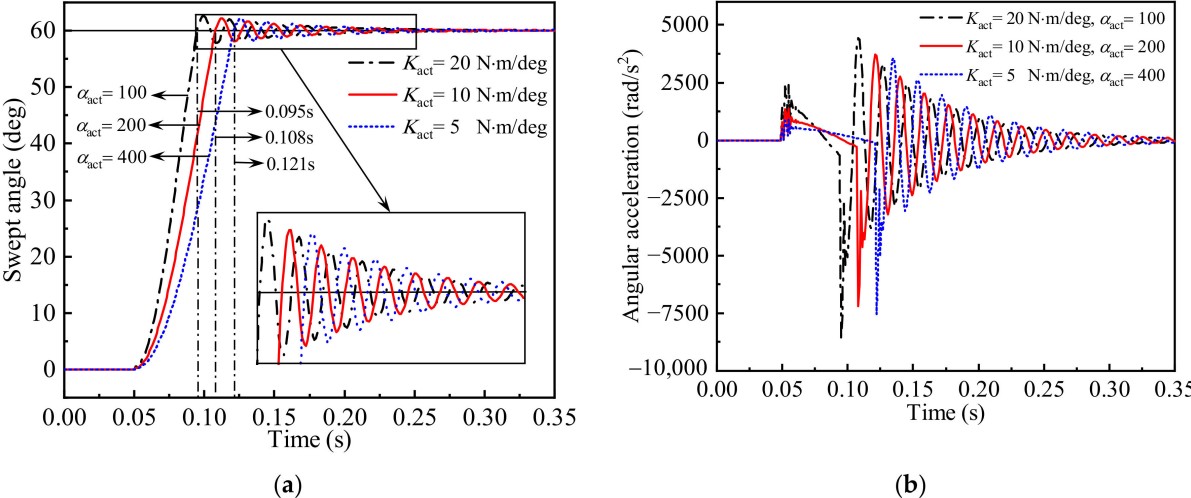

**Figure 13.** Time histories of the overall rotation motion of the wing. (**a**) Wing rotation angle vs. time. (**b**) Angular acceleration vs. time.

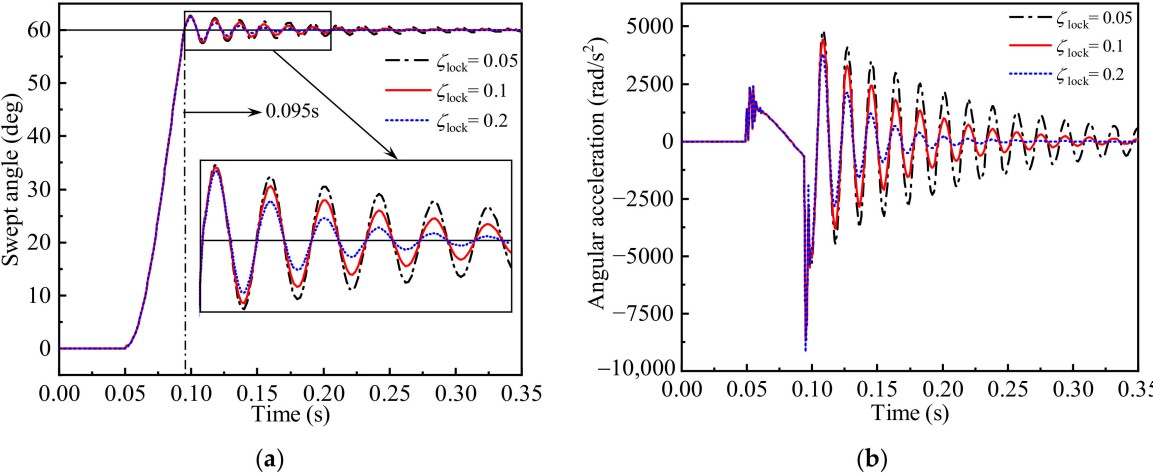

**Figure 14.** Time histories of the rotation motion of the wing. (**a**) Wing rotation angle vs. time. (**b**) Angular acceleration vs. time.

In order to explore the effect of the rotational spring constant mounted at the wing root on the transient responses of the wing, three actuator spring constants, 20 N · m/deg, 10 N · m/deg and 5 N · m/deg, are used for the contrastive analysis. Meanwhile, the stiffness of the locking spring remains fixed, that is $\alpha_{act} K_{act} = 2000$ N · m/deg. The damping ratios are taken as $\zeta_{act} = 0.05$, and $\zeta_{lock} = 0.05$. All the displacements in the figures are expressed in floating frame $ox_w y_w z_w$. It can be seen from Figures 15 and 16 that, in addition to a small fluctuation in the initial variation stage, the chord-wise and z-direction displacements at the wing-tip (see point a in Figure 9a) mainly occur in the post-lock stage, because the wing suffers a large impact in this stage. In addition, because the wing rotates around the z-axis, the chord-wise displacement is much larger than the out-of-plane z-direction displacement in the post-lock stage. The wing-tip torsional angle responses calculated from points a and b in Figure 9a are shown in Figure 17. As can be seen, the torsional vibrations mainly occur in the post-lock stage, at which the wing is subjected to a large rotational moment from the locking spring, as shown in Figure 18. Similarly, compared with the responses at the initial stage of the wing rotation, the incremental internal loads at the wing root, dominated by inertial force and aerodynamic force, are much larger in the post-lock stage, as shown in Figures 19 and 20.

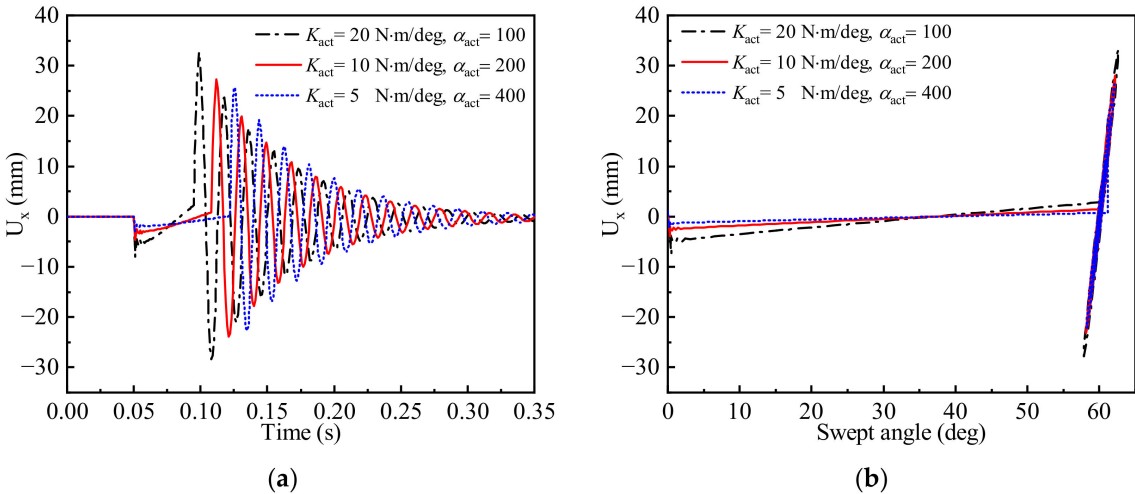

**Figure 15.** Transient responses of the chord-wise displacement at the wing tip. (**a**) Displacement vs. time. (**b**) Displacement vs. sweep angle.

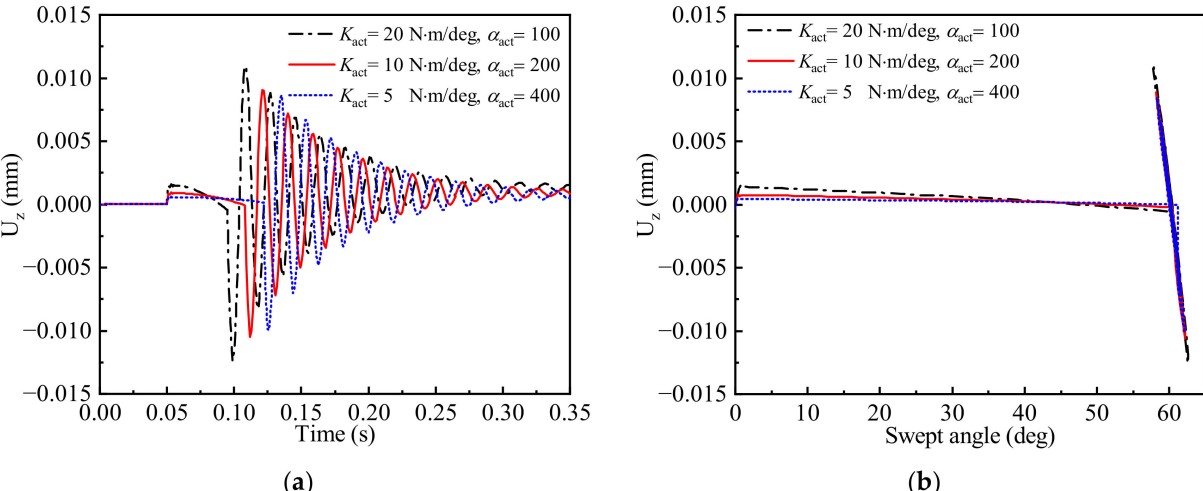

**Figure 16.** Transient responses of the displacement in z direction at the wing tip. (**a**) Displacement vs. time. (**b**) Displacement vs. sweep angle.

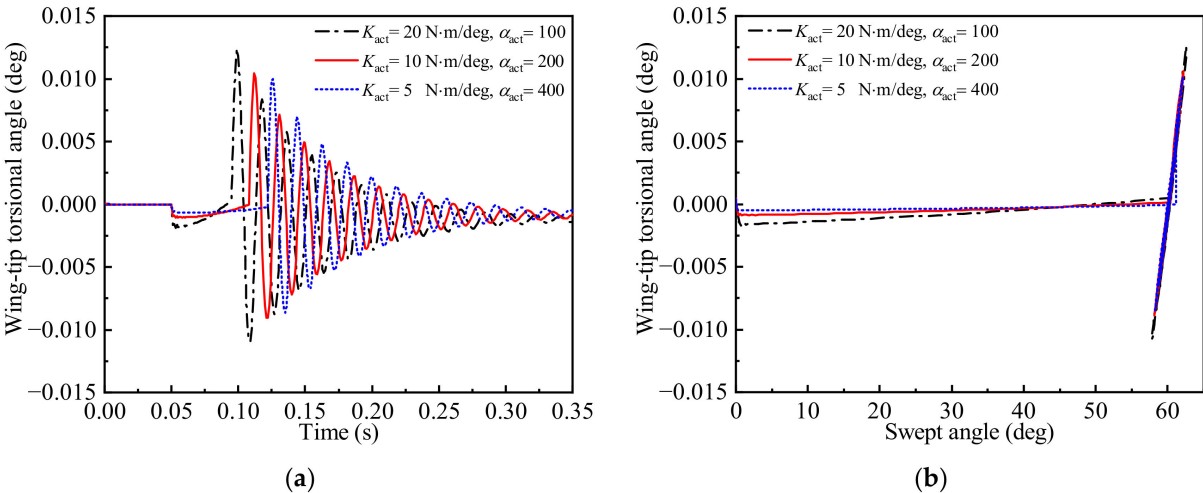

**Figure 17.** Transient responses of the wing-tip torsional angle. (**a**) Torsional angle vs. time. (**b**) Torsional angle vs. sweep angle.

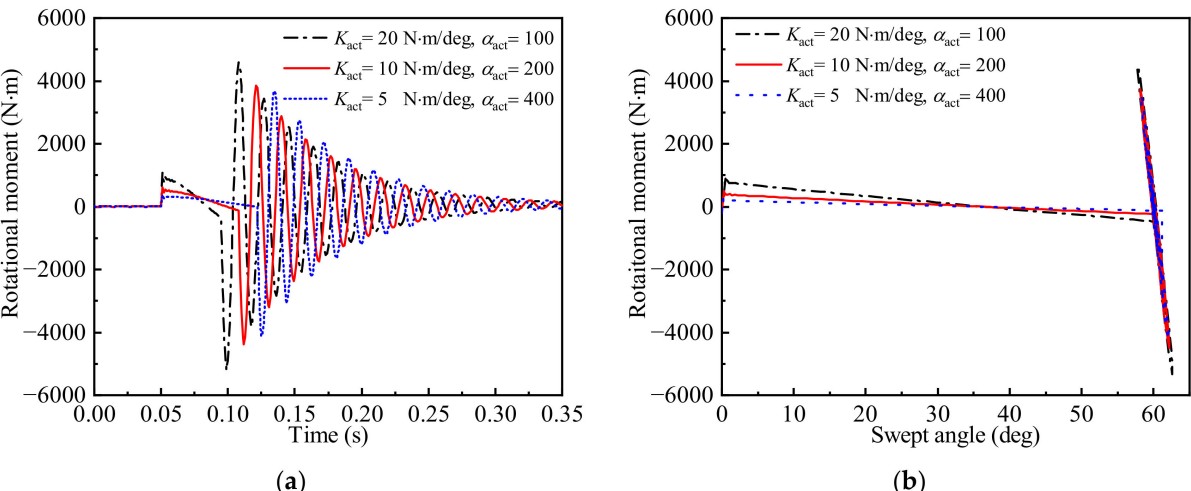

**Figure 18.** Transient responses of the wing root rotational moment. (**a**) Rotational moment vs. time. (**b**) Rotational moment vs. sweep angle.

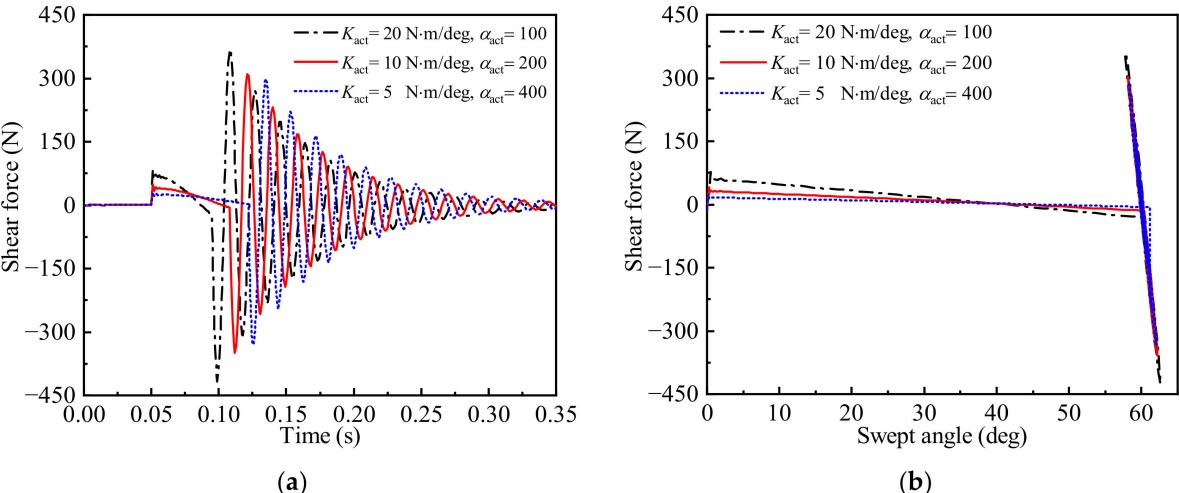

**Figure 19.** Transient responses of the wing root shear force. (**a**) Shear force vs. time. (**b**) Shear force vs. sweep angle.

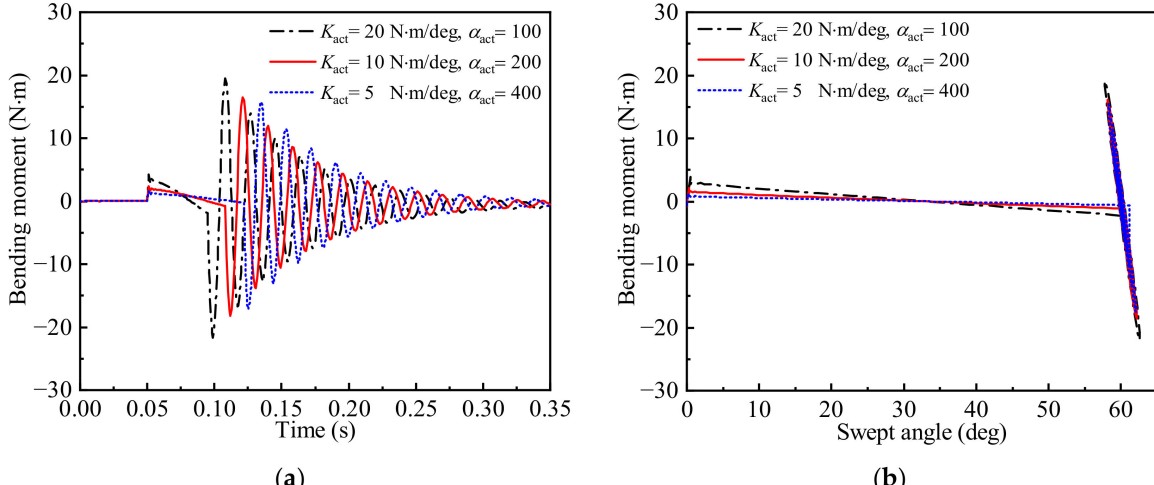

**Figure 20.** Transient responses of the wing root bending moment. (**a**) Bending moment vs. time. (**b**) Bending moment vs. sweep angle.

Figures 21–24 show the time histories of the rotation moment, wing-root shear force, wing-root bending moment, and wing-tip torsional angle, respectively. Obviously, the large stiffness of the post-lock spring produces large torsional moment. The amplitudes of the responses of the wing-root shear force, bending moment and the wing-tip torsional angle increase with the increase of the stiffness of the locking spring. Therefore, from the perspective of the structural strength design, it is necessary to reasonably determine the stiffness of the locking spring.

Finally, the influence of the quadratic velocity term $F_{sv}(q, \dot{q})$ on the transient displacement responses of the wing during the rapid morphing is investigated. Two sets of the actuator spring constant, $K_{act} = 20 \, \text{N} \cdot \text{m/deg}$ and $K_{act} = 60 \, \text{N} \cdot \text{m/deg}$, are used for comparisons. The simulation results are given by Figures 25 and 26, respectively. In fact, the quadratic velocity vector in Equation (20) reflects the comprehensive influence including the Coriolis force and centrifugal force. For the present model, the influence of the Coriolis force can be neglected. It can be seen from Figures 25 and 26 that, at the initial stage of wing rotation, there is a small out-of-plane displacement at the wing tip. Due to the effect of centrifugal force, the quadratic velocity term decreases the deviation of out-of-plane displacement form that produced at the initial stage of wing rotation. However, overall, the influence of the quadratic velocity term on the out-of-plane bending deformation is very small in the present simulations. In the post-lock stage, this effect can be completely neglected.

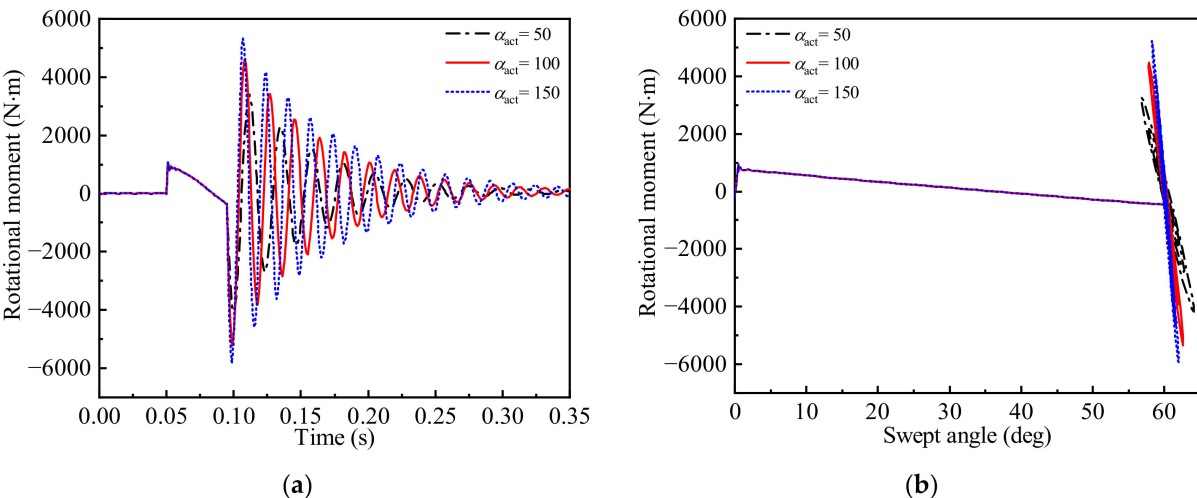

**Figure 21.** Transient responses of the rotational moment. (**a**) Rotational moment vs. time. (**b**) Rotational moment vs. sweep angle.

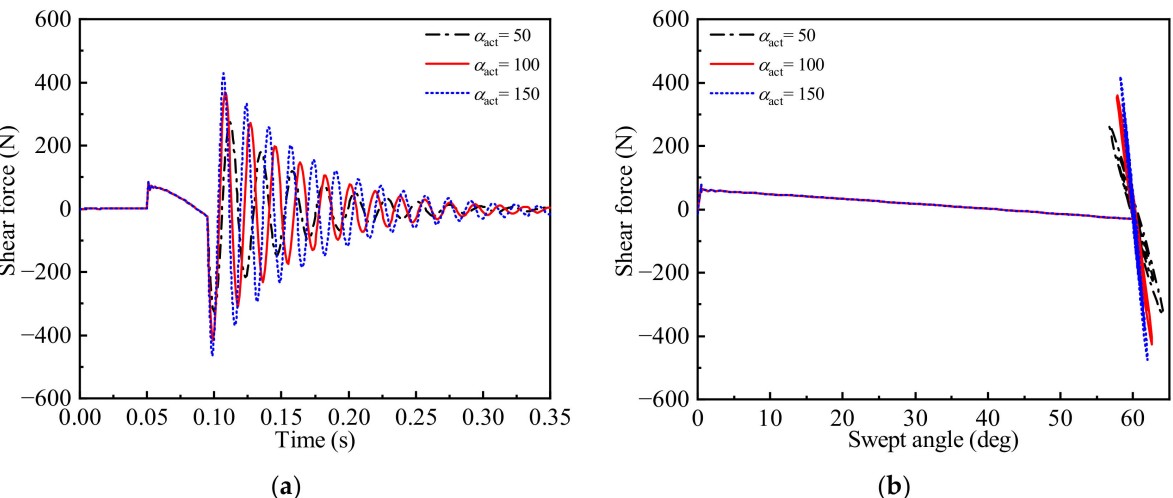

**Figure 22.** Transient responses of the wing-root shear force. (**a**) Shear force vs. time. (**b**) Shear force vs. sweep angle.

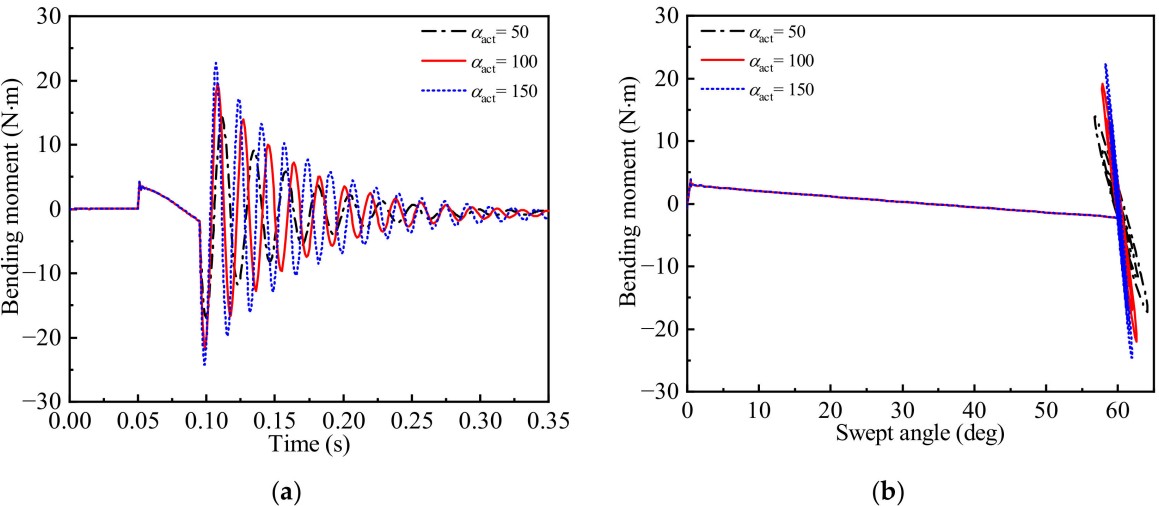

**Figure 23.** Transient responses of the wing-root bending moment. (**a**) Bending moment vs. time. (**b**) Bending moment vs. sweep angle.

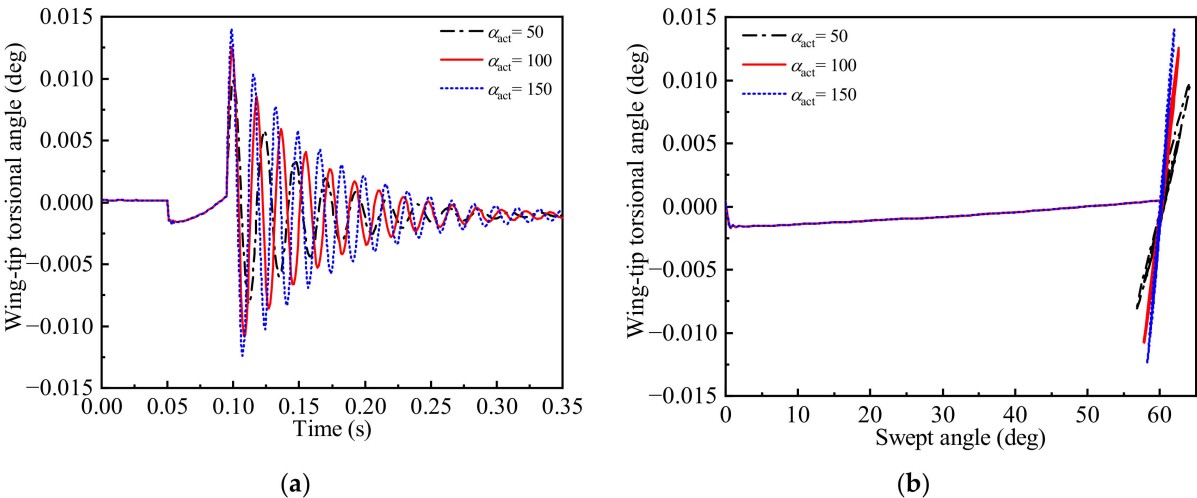

**Figure 24.** Transient responses of the wing-tip torsional angle. (**a**) Torsional angle vs. time. (**b**) Torsional angle vs. sweep angle.

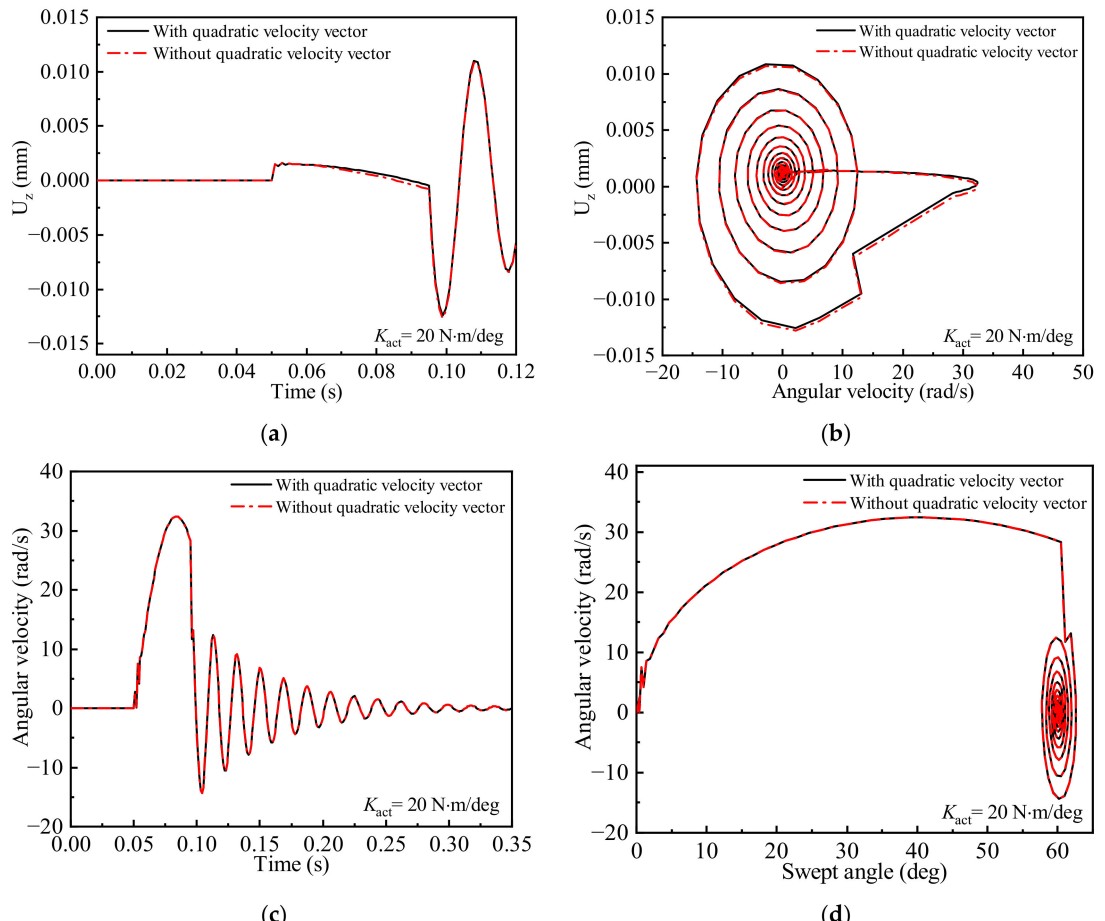

**Figure 25.** Effect of the quadratic velocity term on the displacement responses, $K_{act} = 20 \, \mathrm{N \cdot m/deg}$. (**a**) Z-displacement vs. time. (**b**) Z-displacement vs. angular velocity. (**c**) Angular velocity vs. time. (**d**) Angular velocity vs. sweep angle.

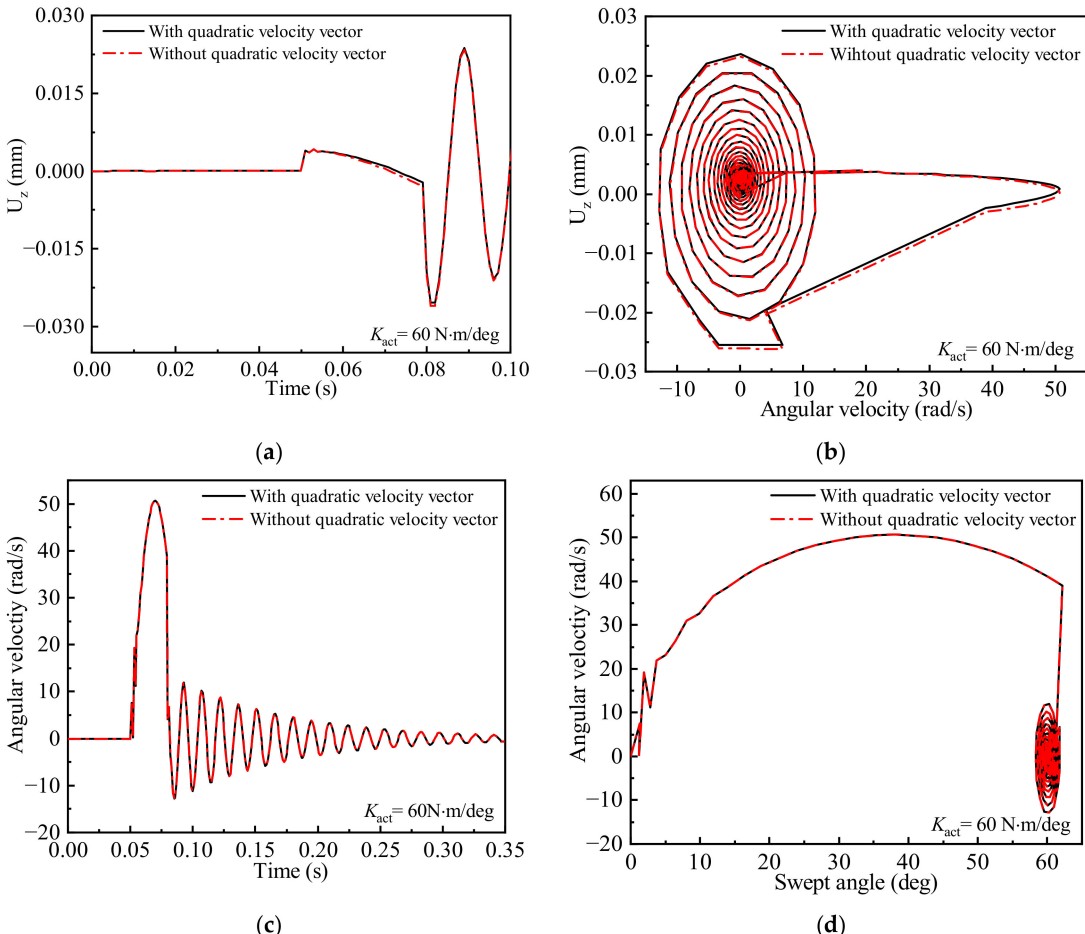

**Figure 26.** Effect of the quadratic velocity term on the displacement responses, $K_{act} = 60\,\mathrm{N} \cdot \mathrm{m/deg}$.
(**a**) Z-displacement vs. time. (**b**) Z-displacement vs. angular velocity. (**c**) Angular velocity vs. time.
(**d**) Angular velocity vs. sweep angle.

## 4. Conclusions

In order to predict the transient aeroelastic responses of a variable swept missile wing
during the rapid morphing process, in this paper, a computationally efficient time-varying
aeroelastic modeling method was developed. The finite element model generated by the
commercial software platform was combined with the floating frame method to describe
the rigid–flexible coupling dynamic characteristics of the rotating wing. By integrating the
structural dynamics model with the piston-based aerodynamic model, a set of time-varying
aeroelastic equations was established. It is pointed out that the transient aeroelastic analysis
of the rotating wing requires the real-time calculation of the time-varying lifting surface.
The flutter analysis under different sweepback angles shows that the flutter characteristics
of the wing are greatly affected by the sweep angle. With the increase of the sweep angle,
there exists a jumping phenomenon in the flutter speed due to the flutter mode switching,
which requires special attention in the aeroelastic design.

Transient aeroelastic responses were predicted under various parameters, such as the
actuator and locking spring constants, and the damping ratio in the rotational degree of
freedom. It is demonstrated that the responses at the initial stage of the wing rotation are
small, and the relatively large responses mainly occur in the post-lock stage. Therefore, it is
necessary to suppress vibrations by reasonably designing the stiffness and damping in the
post-lock stage.

**Author Contributions:** Conceptualization, Y.Z.; methodology, Y.Z. and L.Z.; software, L.Z.; formal analysis, L.Z.; investigation, L.Z.; resources, L.Z.; data curation, L.Z.; writing—original draft preparation, L.Z.; writing—review and editing, Y.Z.; All authors have read and agreed to the published version of the manuscript.

**Funding:** The research was funded by the National Natural Science Foundation of China, Grant No.11472128.

**Data Availability Statement:** All data used during the study appear in the submitted article.

**Acknowledgments:** The study described in the paper was supported by the National Natural Science Foundation of China (Grant No.11472128).

**Conflicts of Interest:** The authors certify that there are no conflict of interest with any individual/organization for the present work.

## Nomenclature

| | |
|---|---|
| $A$ | the transformation matrix |
| $C_{\text{act}}$ | the damping coefficients in the rotation |
| $C_{\text{lock}}$ | the damping coefficients in the locking phase |
| $C_{qq}$ | modal damping matrix |
| $c_d$ | the drag coefficient |
| $F_{\text{const}}$ | the constant moment |
| $F_{sv}(q, \dot{q})$ | the quadratic velocity vector |
| $F_a$ | the generalized unsteady aerodynamic forces (GAF) |
| $f_{ap}$ | the aerodynamic force vector |
| $f_{lp}$ | the lift force |
| $f_{dp}$ | the drag force |
| $f_{cr}(\theta)$ | the correction factor |
| $G_{as}, G_{\alpha s}$ | the spline matrix |
| $H(x, y)$ | the airfoil thickness |
| $h_a$ | the displacement vector at the interpolation points |
| $h_\alpha$ | the slope vector at the interpolation points |
| $K_s$ | the torsional spring constant |
| $K_{qq}$ | the modal stiffness matrix |
| $K_{\text{act}}$ | the nominal stiffness coefficient |
| $m_p$ | the lumped mass |
| $M_w(q)$ | the time-varying mass matrix |
| $M_\infty$ | the Mach number |
| $n_{sl}$ | the number of the finite element nodes |
| $n_{sd}$ | the number of the finite element nodes |
| $q_w(t)$ | the modal coordinate vector |
| $Q_{al}$ | the modal aerodynamic force |
| $S_{wp}$ | the modal shapes |
| $\widetilde{S}$ | the skew symmetric matrix |
| $S_{wp}$ | the vibration mode |
| $T_w$ | the kinetic energy |
| $\bar{u}_{wp}$ | the position vector of the node |
| $U_{\text{act}}$ | the elastic potential energy |
| $U_s$ | the strain energy |
| $u$ | the global FEM displacement vector |
| $\theta_{\text{preload}}$ | the preloaded spring angle |
| $\theta_{\text{max}}$ | the max rotation angle |
| $\theta$ | the sweep angle |
| $\alpha$ | the static angle of attack |
| $\Lambda$ | the leading edge sweep angle |

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
