# Peer review of "Time-Varying Aeroelastic Modeling and Analysis of a Rapidly Morphing Wing"

_aerospace, doi:10.3390/aerospace10020197_

Round 1

Reviewer 1 Report

The work is interesting and relevant, a method for aeroelastic modeling of a variable airfoil wing has been developed. It is shown that the proposed modeling method is computationally efficient and can be applied to both slow and fast time-varying processes of a variable-swept wing. The effectiveness of the proposed method was verified by numerical simulation of the variable wing of a specific aerodynamic model. Flutter characteristics at different swept angles were determined. The flutter analysis under different sweepback angles shows that the flutter charac teristics of the wing are greatly affected by the swept angle. The work is fully consistent with the subject of the journal and can be published.

Author Response

Response to Reviewer 1 Comments

The work is interesting and relevant, a method for aeroelastic modeling of a variable airfoil wing has been developed. It is shown that the proposed modeling method is computationally efficient and can be applied to both slow and fast time-varying processes of a variable-swept wing. The effectiveness of the proposed method was verified by numerical simulation of the variable wing of a specific aerodynamic model. Flutter characteristics at different swept angles were determined. The flutter analysis under different sweepback angles shows that the flutter charac teristics of the wing are greatly affected by the swept angle. The work is fully consistent with the subject of the journal and can be published.

Response : Thank you for your important approval. The team always insists  that the establishment of an effective time-varying aeroelastic model is the prerequisite for the performing transient response analysis for the rapidly morphing aircraft. This paper includes an important result of the hard work over the whole team recently. Therefore, your recognition is a great encouragement for the team. Tanks you very much sincerely.

Reviewer 2 Report

The term "aeroelasticity" refers to the interaction between aerodynamic forces and structural
deformations in an aircraft, leading to dynamic behaviour such as flutter or divergence. Time-
varying aeroelastic modelling involves the study of these interactions in the presence of changes
in wing geometry over time, such as in the case of a rapidly morphing wing. The goal of this
analysis is to understand the potential aerodynamic and structural responses of the wing under
different operating conditions and to ensure safe and efficient performance.

The analysis may involve numerical simulations, wind tunnel tests, or flight tests to validate the
model and to make predictions about the wing's behaviour.

In the current manuscript, authors have utilized the numerical techniques to predict the behaviour
of wings behaviour.

The most important aspect of this manuscript which is one of the strengths of it is that the analysis
has involved structural analysis, FEM based modelling and nonlinear dynamical approaches are
connected together to provide a method for real-time determination of time varying lifting surface
during morphing phenomenon.

After going through the details of the manuscript, my confidence in this article is positive,
however, the room for improvement is always open and for that I would like to give some
suggestions before it can be proceeded for final publication.

General Comments

1. As this article is based on nonlinear analysis and a lot of mathematical equations are part
of it, therefore I will encourage authors to add nomenclature to it. A lot of times, things are
clear to authors but from readers point of view it is important to add nomenclature for better
understanding and enhancing readability of work

2. The use of articles the, a, an etc are missing at some places, please go through the
manuscript again and remove any grammatical or language issues present in article.

3. Please refer to the source of the equations where possible if they are from some reference

Author Response

Dear Reviewer

Thank you

Reviewer 3 Report

This article is about morphing wings. This topic has been very relevant and fashionable in recent years. The authors used their own model based on the dynamic structural model combined with the supersonic piston theory. The model was verified by comparing the results with commercial software. They then tested the aeroelastic responses for three different actuator stiffness and actuator spring constant. In the end, the time histories of investigated parameters were presented.

The paper was written carefully and carefully. Correct language. I found no inaccuracies in the reasoning and the data was presented very clearly. Only figures 11 and 12 have unclear markings. What authors understood as 1st, 2nd, etc.? This should be explained in the text.

A very carefully prepared article. Congratulations.

Author Response

Response to Reviewer 3 Comments

This article is about morphing wings. This topic has been very relevant and fashionable in recent years. The authors used their own model based on the dynamic structural model combined with the supersonic piston theory. The model was verified by comparing the results with commercial software. They then tested the aeroelastic responses for three different actuator stiffness and actuator spring constant. In the end, the time histories of investigated parameters were presented.

Point : The paper was written carefully and carefully. Correct language. I found no inaccuracies in the reasoning and the data was presented very clearly. Only figures 11 and 12 have unclear markings. What authors understood as 1st, 2nd, etc.? This should be explained in the text.

Response : Thank you for your valuable comment and recognition. The unclear markings in the figures 11 and 12 have been explained in the text. And your recognition is a great encouragement for the team.

Part of the 3.1 section : In order to analyze the flutter character, Figure 11 and 12 provide structural damping varies with incoming flow speeds for the first six modes (1st, 2nd, 3rd, 4th and 6th). When the swept angle is less than  , the fifth-order mode becomes unstable. In this range, the flutter is characterized by the hump shape, as shown in Figure 11, in which the unsta-ble mode crosses the zero point () twice. Figure 12 shows the flutter curves at the sweep angle of  . It can be seen that with the increase of the flow speed, the fifth-order modal branch reaches the zero point, and then drops rapidly, which indicate that a switch of the unstable mode is coming. When the swept angle increases to  , the third-order modal branch first crosses the zero point and becomes the unstable mode. Therefore, the essential reason for the jumping phenomenon of flutter speed is the switching of the unstable modal branches.
